# Learning interventions and training methods in health emergencies: A scoping review

**Heini Utunen**[1]*, **Giselle Balaciano**[1], **Elham Arabi**[1], **Anna Tokar**[1], **Aphaluck Bhatiasevi**[1], **Jane Noyes**[2]

**1** Health Emergencies Programme, Learning and Capacity Development Unit, World Health Organization, Genève, Switzerland, **2** Department of Medical and Health Sciences, Bangor University, Bangor, United Kingdom

* utunenh@who.int

**Data Availability Statement:** All relevant data are within the paper and its Supporting Information files.

## Abstract

### Background

Keeping the health workforce and the public informed about the latest evolving health information during a health emergency is critical to preventing, detecting and responding to infectious disease outbreaks or other health emergencies. Having a well-informed, ready, willing, and skilled workforce and an informed public can help save lives, reduce diseases and suffering, and minimize socio-economic loss in affected communities and countries. Providing "just in time" support and opportunities for learning in health emergencies is much needed for capacity building. In this paper, 'learning intervention' refers to the provision of ad-hoc, focused, or personalized training sessions with the goal of preparing the health workers for emergencies or filling specific knowledge or skill gaps. We refer to 'training methods' as instructional design strategies used to teach someone the necessary knowledge and skills to perform a task.

### Methods

We conducted a scoping review to map and better understand what learning interventions and training methods have been used in different types of health emergencies and by whom. Studies were identified using six databases (Pubmed/Medline, Embase, Hinari, WorldCat, CABI and Web of Science) and by consulting with experts. Characteristics of studies were mapped and displayed and major topic areas were identified.

### Results

Of the 319 records that were included, contexts most frequently covered were COVID-19, disasters in general, Ebola and wars. Four prominent topic areas were identified: 1) Knowledge acquisition, 2) Emergency plans, 3) Impact of the learning intervention, and 4) Training methods. Much of the evidence was based on observational methods with few trials, which likely reflects the unique context of each health emergency. Evolution of methods was apparent, particularly in virtual learning. Learning during health emergencies appeared to

**Funding:** The authors received no specific funding for this work.

**Competing interests:** The authors have declared that no competing interests exist.

improve knowledge, general management of the situation, quality of life of both trainers and affected population, satisfaction and clinical outcomes.

## Conclusion

This is the first scoping review to map the evidence, which serves as a first step in developing urgently needed global guidance to further improve the quality and reach of learning interventions and training methods in this context.

## Introduction

Learning in health emergencies can provide a foundation for building capacity for emergency preparedness and response specific to different health emergencies [1] (e.g., biological, environmental, armed conflicts, deliberate acts of terrorism, industrial accidents), especially in low- and middle-income countries where health systems need to be further strengthened [2]. A ready, willing, and able workforce is required that can be called upon in health emergencies to help save lives, reduce disease and suffering, and minimize socio-economic loss in affected communities and countries. In today's interconnected landscape, an educated public is also needed to champion measures for a strong emergency preparedness and effective response [1]. This can be achieved by guiding, standardizing and facilitating the delivery of life-saving knowledge, first to frontline workers in health emergencies and second to the public. Learning paves the way for strengthened health literacy and understanding of health communication which, in turn, bolsters awareness and support for measures needed during health emergency events [3].

The World Health Organization (WHO) is developing guidance on *Learning in Emergencies* to provide evidence-based interventions, training methods and tools for professionals, communities and institutions to ensure quality learning provision and knowledge dissemination during health emergencies. The primary reason for developing this guidance emerged from lessons learned from the COVID-19 pandemic, which presented unique circumstances and global challenges. It became clear that 'just in time' learning is required to retrain and upskill large numbers of health professionals in order to launch an effective response. For example, new methods and approaches for testing and surveillance of COVID-19 at scale were proven essential to operate at the national, regional, and local level, which involved the need for rapid dissemination and uptake of new learning so that the entire health, social care and education workforce were equipped with relevant knowledge and skills to respond effectively, especially in low- and middle-income countries [4, 5].

The COVID-19 pandemic also prompted the need to integrate effective risk communication strategies to tackle the evolving information needs and misinformation or infodemic (i.e., short for information epidemic) [6] and potential spread of misinformation that could negatively impact on efforts to develop and deliver learning interventions to professionals, communities and institutions [7]. From the onset of the COVID-19 pandemic, the World Health Organization (WHO) committed to knowledge dissemination to support frontline health workers, governmental and non-governmental actors, policy makers, capacity builders and trainers as well as the public via its low-bandwidth adjusted online platform. In addition to regular updates as more research emerged, WHO intended to regularly assess the effectiveness of this initiative—that is knowledge dissemination through its online platform (OpenWHO) on all the aspects of tackling the pandemic and then including other health emergencies. Key

findings indicated that employing strategies to make learning more equitable and accessible can yield better results in terms of outreach, and that learning production must be targeted for real-time events in languages spoken in outbreak impacted areas [8]. Nonetheless, learning design based on inclusive pedagogy and learning sciences, while being cognizant of barriers to accessing learning (such as poor internet connection or limited digital literacy) can optimize learning experience [9].

The second rationale for developing the Learning in Emergencies guidance is that various current guidelines tackled adult learning partially or in one dimension. WHO itself has published several frameworks and recommendations. However, none of these related guidelines covers Adult Learning in Health Emergencies. The Learning in Emergencies guidance will address a priority issue cited during the World Health Assembly (WHA), to help WHO support governments in their health-related capacity building and to reach their health learning goals. The *Learning in Emergencies* guidance will encompass the full scope of learning as preparedness, readiness, response and resilience actions in capacitation for public health emergencies.

This scoping review of literature was undertaken to inform further methodological choices and the commissioning of subsequent systematic reviews to feed into the Guidance development process.

## Objectives

The objective of this scoping review was to map the existing evidence (including qualitative, quantitative and mixed-methods peer-reviewed publications, systematic and other types of literature reviews and qualitative evidence synthesis) that have been published on the topic.

## Target population

The target population included experts and individuals in need of health information, such as the health workforce, experts and volunteers, national institutions and ministries (governmental and non-governmental actors), policymakers, academia, capacity builders and trainers, citizens and affected populations.

## Phenomenon of interest

The phenomena of interest were classified as learning interventions as specified below and health emergencies including the follow types: Biological, Environmental, Armed conflicts, Deliberate acts of terrorism and Industrial accidents.

## Intervention/Exposure

Learning interventions included:

- Continuous learning for professionals preparing for or acting in health emergencies

- Adult learning interventions and methods in emergency situations

- Professional education, training and learning

- Real time, just in time learning

- Knowledge and learning transfer from an expert organization

- Learning readiness in anticipation of and preparedness for any health hazard

    Exclusion criteria:

- Professional learning

- University degrees

- Postgraduate studies

## Methods

The methodology was guided by Arksey and O'Malley's [8] five-stage framework for scoping reviews: identifying the research question(s), identifying relevant studies, charting the data, collating, summarizing, and reporting the results. In the latter case this means reporting the results of the searches and mapping the studies and not providing any detailed analysis of included study results. In addition, principles of mixed-methods framework synthesis were used to manage diverse study designs and help extract, map, chart, categorize and summarize studies under four prominent topic areas [10]. Like McGill and colleagues in their recent scoping review on knowledge exchange in crisis settings [11], we did not go beyond Arksey and O'Malley's scoping review methodology to undertake a synthesis of findings. This aligned with the purpose of this scoping review to identify published peer reviewed studies from which subsequent review questions could be formulated and systematic reviews commissioned. The review was reported using the relevant domains of the Preferred Reporting items for Systematic Review and Meta-Analysis for scoping reviews (PRISMA-ScR) [12].

A priori protocol was developed and published on the Open Science Framework: https://doi.org/10.17605/OSF.IO/5BK9R.

### Selecting relevant studies

Searches were carried out between 2 February and 28 February 2023, and covered sources from 2003 to the present. Key search terms were identified within three PICO (Problem/Population, Intervention, Comparison, Outcome) question components (S1 Table in S1 File). These terms were also used to identify relevant documents from which Medical Subject Heading (MeSH) or other database-specific terms and keywords could be extracted.

Key terms were searched using a free text strategy in the titles and abstracts. This allowed having a broader, more sensitive approach and eliminated the possibility of relevant items being missed. MeSH was applied to give more specific results.

The following databases were searched: Pubmed/Medline, Embase, Hinari, WorldCat, CABI and Web of Science, using predefined combinations of key search terms. To prioritize LMICs (low- and middle-income countries), filters of the selected databases were applied. The search strategy and search words are annexed (Full strategy in supplementary material). A systematic grey literature search was performed and, due to the nature and objective of the information retrieved, will be published in a separate article.

Search results were scanned for relevance and those meriting further examination were imported into Rayyan for further consideration. The search and initial screening were undertaken by GB and studies checked by other authors were included. Citations were excluded if they focused exclusively on academic settings (early education through medical school), did not have a learning intervention (for example studies addressing knowledge in unprepared professionals) or were not contextualized during or for a health emergency. News (or announcements) as well as protocols of studies/reviews that had not been completed, and other irrelevant document types were excluded, as well as non-peer reviewed articles.

Titles and abstracts were screened against the following inclusion criteria: (1) published between 2003 and 2023; (2) abstract and title written in English; (3) presenting data on the

research question (date, methodology, focus on geographical low-, middle- and high-income context, type of learning/intervention and for whom, type of emergency); (4) peer-reviewed sources only. Then screening of full texts of pre-selected citations against above-mentioned criteria was performed. At this stage full texts written in other United Nations languages and Portuguese language were included as well. Processes were undertaken by GB and crossed checked by co-authors. We did not undertake double-blind processing.

### Charting the data

All the included evidence was extracted into an Excel spreadsheet and included citations were exported into Endnote. Data were extracted systematically using a standardized form that included information on the period of study, location, study population, design, research questions, key findings, and conclusions.

### Collating, summarizing and reporting results

The charted data (in the form of an evidence map) were then further analyzed, grouped and sorted, guided by the review aim and objectives. We specifically focused on developing an understanding of the available literature on the phenomena of interest and created visual displays and tables. We also identified and described four major topic areas.

## Results

A total of 6411 articles were imported into Rayyan, of which 1656 were duplicates and 4317 were excluded based on a brief scan of titles and abstracts. The remaining 446 documents were sorted by full-text access. One hundred and thirteen articles were excluded for not being relevant and fourteen did not have full text access. As shown in Fig 1, 319 articles met the inclusion criteria and were included. Full description of included studies can be found in Table 1.

### Overview of study characteristics

Most of the articles were descriptive studies on experiences in different countries [13–177], followed by before-after studies [170, 174, 178–261]. Also, we found 19 randomized controlled trials [262–280], 14 cross-sectional studies [281–294], 12 observational studies [295–306], 7 reviews [307–313], 5 qualitative studies [314–318] and 3 opinion pieces [319–321]. (S2 Table in S1 File)

### Emergency context

COVID-19, disasters in general, Ebola and wars were the most frequent topics. This is related to the publication dates because most studies have been published during these health emergencies (Ebola, 2015 and COVID-19 during 2020, 2021 and 2022) (Fig 2)

### Location

Evidence was found in 67 countries and regions: most studies were conducted in China (n = 36), mainly evidence about COVID-19 and earthquakes, India (n = 19) and South Africa (n = 15). A large number of the learning evidence was also derived from several countries in Africa (n = 25) and also international settings (n = 23).

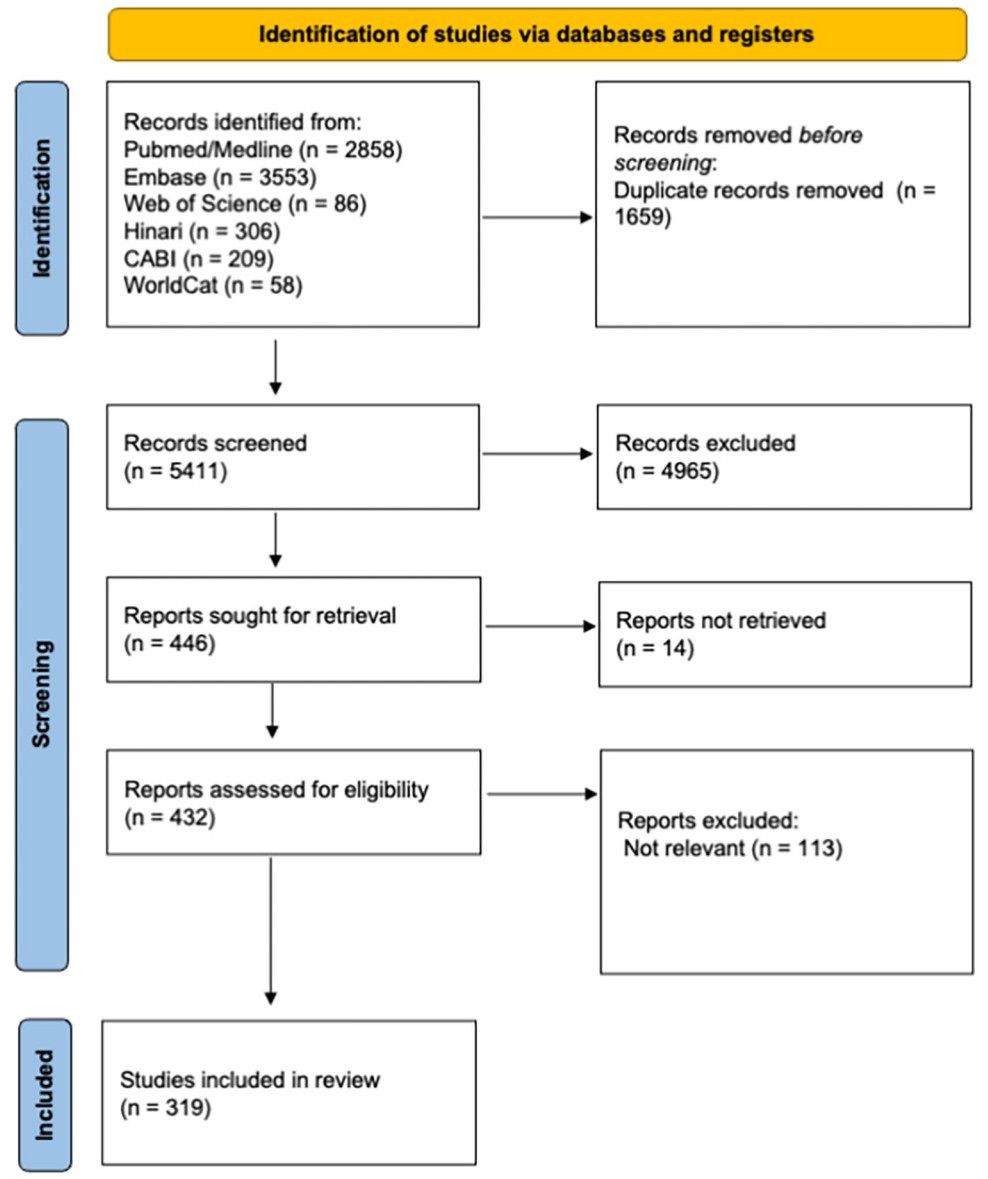

**Fig 1. PRISMA flowchart.**

## Interventions, learning methods and tools

Different types of learning interventions are described across the studies. The majority of the studies used in-person training as an intervention [13, 15, 19–23, 31, 34, 35, 38–43, 47, 49–51, 53, 56, 62, 67, 68, 71, 73, 74, 76–79, 81–83, 88, 94, 98–101, 105, 109, 110, 112, 119, 121, 123–127, 129–132, 134, 136, 137, 139–142, 144, 145, 147–152, 154–160, 162, 166, 173, 175, 179–183, 185, 186, 188–190, 194, 196, 198, 199, 201, 207, 211, 213, 215–218, 221, 223, 224, 226, 229, 230, 233, 235, 238, 239, 244–246, 248–255, 258, 260, 262, 263, 265, 267, 269, 270, 272, 274, 276, 278, 280, 283, 289–291, 293, 295, 296, 299, 303, 314, 316, 318, 319, 321, 322], followed by virtual training.

Lectures, discussions, role-playing, 'hands-on' basic skills training, materials, videos and simulations were used as in-person modalities. Additionally, 9 studies [27, 84, 86, 90, 118, 225,

**Table 1. Description of included studies.**

| Author | Year | Title | Country | Disaster/ Emergency | Population | Learning Method | Major topic | Study design |
|---|---|---|---|---|---|---|---|---|
| Panda R | 2022 | Evaluation of COVID-19 ECHO training program for healthcare workers in India—A Mixed-Method Study. | India | COVID-19 | Health workforce | Virtual training | Knowledge evaluation | Cross-sectional |
| Camputaro LA | 2021 | Intensive competency-based training strategy in a National Hospital in times of Pandemic. | El Salvador | COVID-19 | Health workforce | In-person training | Knowledge evaluation | Qualitative |
| Siddiqui SS; | 2023 | The impact of a "short-term" basic intensive care training program on the knowledge of nonintensivist doctors during the COVID-19 pandemic: An experience from a population-dense low- and middle-income country. | India | COVID-19 | Health workforce | In-person training | Knowledge evaluation | Observational |
| Jordan P | 2023 | Development of a training programme for professional nurses in South Africa—An educational response to the COVID-19 pandemic. | South Africa | COVID-19 | Health workforce | In-person training | Learning technique | Report |
| Kharel R | 2022 | Training program for female community volunteers to combat COVID 19 in rural Nepal. | Nepal | COVID-19 | Experts and volunteers | Virtual training | Learning technique | Report |
| Caviglia M | 2022 | Response to Mass-Casualty Incidents and Outbreaks: A Prehospital Disaster Training Package Developed for the National Emergency Medical Service in Sierra Leone. | Sierra Leone | Disasters | Health workforce | In-person training | Learning technique | Report |
| Singh SS | 2022 | Training community health workers for the COVID-19 response, India. | India | COVID-19 | Health workforce | In-person training | Impact | Cross-sectional |
| Guragai M | 2020 | Medical Students' Response to the COVID-19 Pandemic: Experience and Recommendations from Five Countries. | Brazil, Nepal, the Philippines, Rwanda, and the United States. | COVID-19 | Health workforce and community | Blended | Learning technique | Report |
| Marsh RH; | 2021 | Facing COVID-19 in Liberia: Adaptations of the Resilient and Responsive Health Systems Initiative. | Liberia | COVID-19 | Health workforce | Virtual training | Learning technique | Report |
| Uttekar S | 2023 | Empowering Health Workers to Build Public Trust in Vaccination: Experience from the International Pediatric Association's Online Vaccine Trust Course, 2020–2021. | International | COVID-19 | Health workforce | Virtual training | Knowledge evaluation | Report |
| Zerdo Z; | 2022 | Implementation of a malaria prevention education intervention in Southern Ethiopia: a qualitative evaluation. | Etihopia | others | Health workforce | In-person training | Impact | Randomized controlled trial (RCT) |
| Perera N | 2022 | Implementation of a coronavirus disease 2019 infection prevention and control training program in a low-middle income country. | Syria | COVID-19 | Health workforce | In-person training | Knowledge evaluation | Report |
| Cuen AJ | 2022 | Fighting COVID-19 and HIV through community mobilisation: lessons from an integrated approach to the Africa CDC Partnership to Accelerate COVID-19 Testing (PACT) initiative in seven countries. | Africa | COVID-19 | Health workforce and community | Blended | Learning technique | Review |

*(Continued)*

**Table 1.** (Continued)

| Author | Year | Title | Country | Disaster/ Emergency | Population | Learning Method | Major topic | Study design |
|---|---|---|---|---|---|---|---|---|
| Malik JA; | 2022 | Myths and misconception of COVID-19 among hospital sanitary workers in Pakistan: Efficacy of a training program intervention. | Pakistan | COVID-19 | Health workforce | Virtual training | Knowledge evaluation | Before-after |
| Osula VO | 2022 | COVID-19 advanced respiratory care educational training program for healthcare workers in Lesotho: an observational study. | Lesotho | COVID-19 | Health workforce | In-person training | Knowledge evaluation | Before-after |
| Cianelli R | 2013 | Mental health training experiences among Haitian healthcare workers post-earthquake 2010. | Haiti | Earthquake | Health workforce | In-person training | Impact | Observational |
| Lu Y | 2016 | Chinese military medical teams in the Ebola outbreak of Sierra Leone. | Sierra Leone | Ebola | Military | In-person training | Learning technique | Report |
| Yilmaz | 2021 | Real time COVID-19 online learning for medical students: a massive open online course evaluation. | Turkey | COVID-19 | Health workforce | Virtual training | Impact | Cross-sectional |
| Pek JH | 2020 | Teaching Disaster Site Medical Support in Indonesia. | Indonesia | Disasters | Health workforce | In-person training | Learning technique | Report |
| He LX | 2022 | Perspectives of nursing directors on emergency nurse deployment during the pandemic of COVID-19: A nationwide cross-sectional survey in mainland China | China | COVID-19 | Health workforce | N/A | Emergency plans | Cross-sectional |
| Ray S | 2021 | Innovation in primary health care responses to COVID-19 in Sub-Saharan Africa. | Africa | COVID-19 | Health workforce | N/A | Emergency plans | Review |
| Kochis M; | 2021 | Learning During and From a Crisis: The Student-Led Development of a COVID-19 Curriculum | International | COVID-19 | Health workforce | In-person training | Learning technique | Report |
| Shahrin L | 2022 | In-person training on COVID-19 case management and infection prevention and control: Evaluation of healthcare professionals in Bangladesh. | Bangladesh | COVID-19 | Health workforce | In-person training | Knowledge evaluation | Cross-sectional |
| Müller SA | 2020 | Implementation of the WHO hand hygiene strategy in Faranah regional hospital, Guinea. | Guinea | Ebola | Health workforce | In-person training | Knowledge evaluation | Before-after |
| Fredricks K; | 2017 | Community Health Workers and Disasters: Lessons Learned from the 2015 Earthquake in Nepal. | Nepal | Earthquake | Health workforce | N/A | Emergency plans | Qualitative |
| Eardley W | 2016 | Education and Ebola: initiating the cascade of emergency healthcare training. | Africa | Ebola | Health workforce | In-person training | Knowledge evaluation | Before-after |
| Chemali Z; | 2017 | Humanitarian space and well-being: effectiveness of training on a psychosocial intervention for host community-refugee interaction. | Lebanon | War | Experts and volunteers | In-person training | Impact | Before-after |
| Jaguga F | 2020 | Mental health response to the COVID-19 pandemic in Kenya: a review. | Kenya | COVID-19 | Health workforce and community | Virtual training | Emergency plans | Review |
| Ntahobakurira I; | 2011 | The Rwanda Field Epidemiology and Laboratory Training | Rwanda | others | Health workforce | In-person training | Learning technique | Report |
| Soeters HM | 2018 | Infection prevention and control training and capacity building during the Ebola epidemic in Guinea. | Guinea | Ebola | Health workforce | In-person training | Knowledge evaluation | Before-after |

*(Continued)*

**Table 1.** (Continued)

| Author | Year | Title | Country | Disaster/ Emergency | Population | Learning Method | Major topic | Study design |
|---|---|---|---|---|---|---|---|---|
| Stander M; | 2011 | Hospital disaster planning in the Western cape, South Africa. | South Africa | Disasters | Health workforce | N/A | Emergency plans | Cross-sectional |
| Naghavi Alhosseini | 2018 | Earthquake in the city: using real life gamification model for teaching professional commitment in high school students. | Iran | Earthquake | Citizens and affected population | Virtual training | Emergency plans | Before-after |
| Setiawan E | 2021 | Evaluating knowledge and skill in surgery clerkship during covid 19 pandemics: A single-center experience in Indonesia. | Indonesia | COVID-19 | Health workforce | In-person training | Knowledge evaluation | Cross-sectional |
| Dababnah S | 2019 | Feasibility of a trauma-informed parent-teacher cooperative training program for Syrian refugee children with autism. | Turkey | War | Citizens and affected population | In-person training | Impact | Before-after |
| Hawkes M; | 2009 | Use and limitations of malaria rapid diagnostic testing by community health workers in war-torn Democratic Republic of Congo. | Congo | others | Health workforce | In-person training | Knowledge evaluation | Before-after |
| Bustamante ND | 2020 | The Haiti Humanitarian Response Course: A Novel Approach to Local Responder Training in International Humanitarian Response. | Haiti | Disasters | Health workforce | Virtual training | Learning technique | Report |
| Wuthisuthimethawee P | 2022 | How the ARCH Project Could Contribute to Strengthening ASEAN Regional Capacities on Disaster Health Management (DHM). | ASEAN Member States | Disasters | Health workforce | Simulation | Learning technique | Report |
| Burlew R | 2014 | Assessing the relevance, efficiency, and sustainability of HIV/AIDS in-service training in Nigeria. | Nigeria | others | Health workforce | Virtual training | Emergency plans | Report |
| Thomas JJ; | 2022 | Participatory Workshop-Based Intervention for Better Preparedness and Awareness About Disaster Management Among Accredited Social Health Activists in India: A Brief Report. | India | Disasters | Health workforce | Simulation | Knowledge evaluation | Before-after |
| Joseph JK | 2012 | Lay health workers and HIV care in rural Lesotho: a report from the field. | Lesotho | others | Health workforce | In-person training | Knowledge evaluation | Cross-sectional |
| Patel U; | 2015 | Ebola Outbreak in Nigeria: Increasing Ebola Knowledge of Volunteer Health Advisors. | Nigeria | Ebola | Experts and volunteers | In-person training | Knowledge evaluation | Before-after |
| Çiçek A | 2020 | Combat medic course: evaluation of trainees' perception of learning and academic-self perception. | Turkey | War | Health workforce | In-person training | Impact | Before-after |
| Sonenthal PD | 2022 | Applying the WHO course to train emergency and inpatient healthcare workers in Sierra Leone early in the COVID-19 outbreak. | Sierra Leone | COVID-19 | Health workforce | In-person training | Knowledge evaluation | Before-after |
| Macfarlane C; | 2006 | Training of disaster managers at a masters degree level: from emergency care to managerial control. | South Africa | Disasters | Health workforce | Academic training | Learning technique | Report |
| Kang HM | 2021 | Development of a Medical Support Training Program for Disaster Management in Indonesia: A Hospital Disaster Medical Support Program for Indonesia. | Indonesia | Disasters | Health workforce | Simulation | Learning technique | Report |

(*Continued*)

**Table 1.** (Continued)

| Author | Year | Title | Country | Disaster/ Emergency | Population | Learning Method | Major topic | Study design |
|---|---|---|---|---|---|---|---|---|
| Magaña-Valladares L | 2018 | A MOOC (massive online open course) as an immediate strategy to train health personnel in the cholera outbreak in Mexico. | Mexico | Cholera | Health workforce | Virtual training | Learning technique | Report |
| de Morais Pinto R | 2021 | Analyzing the reach of public health campaigns based on multidimensional aspects: the case of the syphilis epidemic in Brazil. | Brazil | others | Citizens and affected population | N/A | Emergency plans | Before-after |
| Engelbrecht B | 2021 | Prioritizing people and rapid learning in times of crisis: A virtual learning initiative to support health workers during the COVID-19 pandemic. | South Africa | COVID-19 | Health workforce | Virtual training | Learning technique | Report |
| Zhou M | 2020 | Research on the individualized short-term training model of nurses in emergency isolation wards during the outbreak of COVID-19. | China | COVID-19 | Health workforce | Blended | Impact | Before-after |
| Leitch L | 2009 | A case for using biologically-based mental health intervention in post-earthquake china: evaluation of training in the trauma resiliency model. | China | Earthquake | Health workforce | In-person training | Impact | Report |
| Ma D | 2021 | Does theme game-based teaching promote better learning about disaster nursing than scenario simulation: A randomized controlled trial. | China | Disasters | Health workforce | Virtual training | Knowledge evaluation | RCT |
| Lee PH | 2018 | The effectiveness of an on-line training program for improving knowledge of fire prevention and evacuation of healthcare workers: A randomized controlled trial. | China | Disasters | Health workforce | Virtual training | Emergency plans | RCT |
| Davidson PM | 2021 | Global digital social learning as a strategy to promote engagement in the era of COVID-19. | International | COVID-19 | Health workforce | Virtual training | Learning technique | Observational |
| Gul S | 2008 | Multitasking a telemedicine training unit in earthquake disaster response: paraplegic rehabilitation assessment. | Pakistan | Earthquake | Citizens and affected population | Virtual training | Learning technique | Before-after |
| Ng YM | 2020 | Coronavirus disease (COVID-19) prevention: Virtual classroom education for hand hygiene. | Hong Kong | COVID-19 | Health workforce | Virtual training | Learning technique | Report |
| Patel | 2022 | Simulation-based ventilatory training for the caregivers at primary and rural health care workers in Central India for dealing with COVID-19 pandemic: recommendations. | India | COVID-19 | Health workforce | Simulation | Learning technique | Report |
| Mutabaruka E | 2011 | The West Africa Field Epidemiology and Laboratory Training Program, a strategy to improve disease surveillance and epidemic control in West Africa. | Africa | others | Health workforce | In-person training | Learning technique | Opinion |
| Sommerland N | 2020 | Reducing HIV- and TB-Stigma among healthcare co-workers in South Africa: Results of a cluster randomised trial. | South Africa | others | Health workforce | In-person training | Knowledge evaluation | RCT |
| Furkan Dağcioğlu B | 2020 | Social adaptation status of Syrian refugee physicians living in Turkey. | Syria | War | Health workforce | N/A | Impact | Cross-sectional |

(*Continued*)

**Table 1.** (Continued)

| Author | Year | Title | Country | Disaster/ Emergency | Population | Learning Method | Major topic | Study design |
|---|---|---|---|---|---|---|---|---|
| Kuhls DA | 2017 | Basic Disaster Life Support (BDLS) Training Improves First Responder Confidence to Face Mass-Casualty Incidents in Thailand. | Thailand | Disasters | Health workforce | In-person training | Impact | Before-after |
| Feldman M | 2021 | Community health worker knowledge, attitudes and practices towards COVID-19: Learnings from an online cross-sectional survey using a digital health platform, UpSCALE, in Mozambique. | Mozambique | COVID-19 | Health workforce | Virtual training | Knowledge evaluation | Cross-sectional |
| Dunin-Bell O | 2018 | What do They Know? Guidelines and Knowledge Translation for Foreign Health Sector Workers Following Natural Disasters. | International | Disasters | Health workforce | N/A | Emergency plans | Review |
| Gunnlaugsson G | 2019 | Tiny Iceland' preparing for Ebola in a globalized world. | Iceland | Ebola | Health workforce | In-person training | Knowledge evaluation | Qualitative |
| El-Khani A | 2021 | Enhancing Teaching Recovery Techniques (TRT) with Parenting Skills: RCT of TRT + Parenting with Trauma-Affected Syrian Refugees in Lebanon Utilising Remote Training with Implications for Insecure Contexts and COVID-19. | Syria | War | Citizens and affected population | Virtual training | Impact | RCT |
| Liu L | 2012 | Zero Health Worker Infection: Experiences From the China Ebola Treatment Unit During the Ebola Epidemic in Liberia. | Liberia | Ebola | Health workforce | In-person training | Knowledge evaluation | Report |
| James LE | 2020 | Integrating mental health and disaster preparedness in intervention: a randomized controlled trial with earthquake and flood-affected communities in Haiti. | Haiti | Earthquake | Citizens and affected population | In-person training | Impact | RCT |
| Chua | 2008 | Building partnerships to address the HIV epidemic. | Singapore | others | Health workforce | In-person training | Impact | Report |
| Cruz-Vega | 2016 | Experience in training in emergencies, Division of Special Projects in Health, Instituto Mexicano del Seguro Social. | Mexico | Disasters | Health workforce and community | Blended | Emergency plans | Report |
| Van Heng | 2008 | Non-doctors as trauma surgeons? A controlled study of trauma training for non-graduate surgeons in rural Cambodia. | Cambodia | War | Experts and volunteers | In-person training | Emergency plans | Before-after |
| Talisuna AO | 2020 | The COVID-19 pandemic: broad partnerships for the rapid scale up of innovative virtual approaches for capacity building and credible information dissemination in Africa. | Africa | COVID-19 | Health workforce | Virtual training | Learning technique | Report |
| Ren | 2017 | Experiences in disaster-related mental health relief work: An exploratory model for the interprofessional training of psychological relief workers. | China | Earthquake | Health workforce | In-person training | Impact | Qualitative |
| Najafi Ghezeljeh T; | 2019 | Effect of education using the virtual social network on the knowledge and attitude of emergency nurses of disaster preparedness: A quasi-experiment study. | Iran | Disasters | Health workforce | Virtual training | Knowledge evaluation | Before-after |

*(Continued)*

**Table 1.** (Continued)

| Author | Year | Title | Country | Disaster/ Emergency | Population | Learning Method | Major topic | Study design |
|---|---|---|---|---|---|---|---|---|
| Hou | 2018 | Disaster Medicine in China: Present and Future. | China | Disasters | Health workforce | In-person training | Emergency plans | Report |
| McQuilkin | 2017 | Academic Medical Support to the Ebola Virus Disease Outbreak in Liberia. | Africa | Ebola | Health workforce | In-person training | Emergency plans | Report |
| Hemingway-Foday JJ; | 2020 | Lessons Learned from Reinforcing Epidemiologic Surveillance During the 2017 Ebola Outbreak in the Likati District, Democratic Republic of the Congo. | Congo | Ebola | Health workforce | In-person training | Emergency plans | Report |
| Yao K | 2010 | Ensuring the quality of HIV rapid testing in resource-poor countries using a systematic approach to training. | Africa | others | Health workforce | In-person training | Learning technique | Report |
| Bodas | 2022 | Training Package for Emergency Medical Teams Deployed to Disaster Stricken Areas: Has 'TEAMS' Achieved its Goals? | Italy | Disasters | Health workforce | In-person training | Knowledge evaluation | Before-after |
| Bemah P | 2019 | Strengthening healthcare workforce capacity during and post Ebola outbreaks in Liberia: an innovative and effective approach to epidemic preparedness and response. | Liberia | Ebola | Health workforce | In-person training | Knowledge evaluation | Report |
| Oji MO | 2018 | Implementing infection prevention and control capacity building strategies within the context of Ebola outbreak in a "Hard-to-Reach" area of Liberia. | Liberia | Ebola | Health workforce | In-person training | Knowledge evaluation | Report |
| Najarian | 2004 | Disaster intervention: long-term psychosocial benefits in Armenia | Armenia | Earthquake | Health workforce | In-person training | Emergency plans | Opinion |
| Wang | 2021 | The effectiveness of E-learning in continuing medical education for tuberculosis health workers: a quasi-experiment from China. | China | others | Health workforce | Virtual training | Knowledge evaluation | Before-after |
| Limpakarnjanarat | 2007 | Long-term capacity-building in public health emergency preparedness in Thailand—short report. | Thailand | Disasters | Health workforce | N/A | Emergency plans | Report |
| Djalali | 2009 | A fundamental, national, medical disaster management plan: an education-based model. | Iran | Earthquake | Health workforce | In-person training | Knowledge evaluation | Before-after |
| Carlos | 2015 | Hospital preparedness for Ebola virus disease: a training course in the Philippines. | Philippines | Ebola | Health workforce | In-person training | Knowledge evaluation | Before-after |
| Cherian | 2004 | Essential emergency surgical procedures in resource-limited facilities: a WHO workshop in Mongolia. | China | Disasters | Health workforce | Blended | Learning technique | Report |
| Welton-Mitchell | 2018 | An integrated approach to mental health and disaster preparedness: a cluster comparison with earthquake affected communities in Nepal. | Nepal | Earthquake | Citizens and affected population | N/A | Impact | Before-after |
| Wang | 2010 | Improving emergency preparedness capability of rural public health personnel in China. | China | Disasters | Health workforce | In-person training | Knowledge evaluation | Before-after |
| Vijaykumar | 2006 | Psychosocial interventions after tsunami in Tamil Nadu, India. | India | Disasters | Experts and volunteers | N/A | Impact | Report |

(*Continued*)

**Table 1.** (Continued)

| Author | Year | Title | Country | Disaster/ Emergency | Population | Learning Method | Major topic | Study design |
|--------|------|-------|---------|---------------------|------------|-----------------|-------------|--------------|
| Koca | 2020 | The effect of the disaster management training program among nursing students. | Turkey | Disasters | Health workforce | In-person training | Impact | RCT |
| Zhang | 2021 | Effect of virtual reality simulation training on the response capability of public health emergency reserve nurses in China: a quasiexperimental study. | China | COVID-19 | Health workforce | Virtual training | Knowledge evaluation | RCT |
| Rosa | 2021 | A Virtual Coaching Workshop for a Nurse-Led Community-Based Palliative Care Team in Liberia, West Africa, to Promote Staff Well-Being During COVID-19. | Liberia | COVID-19 | Health workforce | Virtual training | Impact | Before-after |
| Rajasingham | 2011 | Cholera prevention training materials for community health workers, Haiti, 2010–2011. | Haiti | Cholera | Health workforce | In-person training | Learning technique | Report |
| Shin YA | 2018 | The Effectiveness of International Non-Governmental Organizations' Response Operations during Public Health Emergency: Lessons Learned from the 2014 Ebola Outbreak in Sierra Leone. | Sierra Leone | Ebola | Health workforce and community | N/A | Learning technique | Report |
| Ripp JA | 2012 | The response of academic medical centers to the 2010 Haiti earthquake: the Mount Sinai School of Medicine experience. | Haiti | Earthquake | Health workforce | In-person training | Knowledge evaluation | Report |
| Maduka | 2015 | Ethical challenges of containing Ebola: the Nigerian experience. | Nigeria | Ebola | Health workforce | In-person training | Emergency plans | Report |
| Olu O | 2018 | What should the African health workforce know about disasters? Proposed competencies for strengthening public health disaster risk management education in Africa. | Africa | Disasters | Health workforce | Academic training | Emergency plans | Review |
| Math | 2006 | Tsunami: psychosocial aspects of Andaman and Nicobar islands. Assessments and intervention in the early phase. | India | Disasters | Health workforce | In-person training | Emergency plans | Report |
| Chamane | 2022 | The effect of a mobile-learning curriculum on improving compliance to quality management guidelines for HIV rapid testing services in rural primary healthcare clinics, KwaZulu-Natal, South Africa: a quasi-experimental study. | South Africa | others | Health workforce | Virtual training | Knowledge evaluation | Before-after |
| Bazeyo | 2013 | Regional approach to building operational level capacity for disaster planning: the case of the Eastern Africa region. | Africa | Disasters | Health workforce | N/A | Emergency plans | Report |
| Yi | 2018 | Developing and implementing a global emergency medicine course: Lessons learned from Rwanda. | Rwanda | Disasters | Health workforce | In-person training | Learning technique | Report |
| Hébert | 2020 | Video as a public health knowledge transfer tool in Burkina Faso: A mixed evaluation comparing three narrative genres. | Burquina Faso | others | Health workforce | Virtual training | Learning technique | Before-after |

(*Continued*)

**Table 1.** (Continued)

| Author | Year | Title | Country | Disaster/Emergency | Population | Learning Method | Major topic | Study design |
|---|---|---|---|---|---|---|---|---|
| Orach | 2013 | Use of the Automated Disaster and Emergency Planning Tool in developing district level public health emergency operating procedures in three East African countries. | Africa | Disasters | National Institutions | Virtual training | Emergency plans | Report |
| Bazeyo | 2015 | Ebola a reality of modern Public Health; need for Surveillance, Preparedness and Response Training for Health Workers and other multidisciplinary teams: a case for Uganda. | Uganda | Ebola | Health workforce and community | Blended | Emergency plans | Report |
| Sharara-Chami | 2020 | In Situ Simulation: An Essential Tool for Safe Preparedness for the COVID-19 Pandemic | Lebanon | COVID-19 | Health workforce | Simulation | Impact | Before-after |
| El-Bahnasawy | 2014 | Selected infectious disease disasters for nursing staff training at Egyptian Eastern Border | Egypt | others | Health workforce | N/A | Knowledge evaluation | Before-after |
| Leow | 2012 | Mass casualty incident training in a resource-limited environment. | Sierra Leone | Disasters | Health workforce | In-person training | Knowledge evaluation | Before-after |
| Olness | 2005 | Training of health care professionals on the special needs of children in the management of disasters: experience in Asia, Africa, and Latin America. | International | Disasters | Health workforce | In-person training | Impact | Report |
| Otu | 2016 | Using a mHealth tutorial application to change knowledge and attitude of frontline health workers to Ebola virus disease in Nigeria: a before-and-after study | Nigeria | Ebola | Health workforce | Virtual training | Knowledge evaluation | Before-after |
| Orach | 2013 | Performance of district disaster management teams after undergoing an operational level planners' training in Uganda. | Uganda | Disasters | Health workforce | N/A | Emergency plans | Report |
| Pérez-Manchón | 2015 | [Telemedicine, a medical social network for humanitarian aid between Spain and Cameroon]. | Cameroon | Disasters | Health workforce | Virtual training | Learning technique | Report |
| Meade | 2007 | A deployable telemedicine capability in support of humanitarian operations. | Africa | Disasters | Health workforce | Virtual training | Learning technique | Report |
| Findyartini | 2021 | Supporting newly graduated medical doctors in managing COVID-19: An evaluation of a Massive Open Online Course in a limited-resource setting. | Indonesia | COVID-19 | Health workforce | Virtual training | Impact | Before-after |
| Hess | 2004 | Development of emergency medical services in Guatemala. | Guatemala | Disasters | Experts and volunteers | N/A | Emergency plans | Report |
| Yamada | 2007 | Interdisciplinary problem-based learning as a method to prepare Micronesia for public health emergencies. | Hawaii | Disasters | Health workforce | N/A | Emergency plans | Report |
| Kizakevich | 2007 | Virtual simulation-enhanced triage training for Iraqi medical personnel. | Iraq | Disasters | Health workforce | In-person training | Learning technique | Report |
| O'Reilly G | 2008 | In the wake of Sri Lanka's tsunami: the health for the south capacity-building project. | Sri Lanka | Disasters | Health workforce | N/A | Emergency plans | Report |

(*Continued*)

**Table 1.** (Continued)

| Author | Year | Title | Country | Disaster/ Emergency | Population | Learning Method | Major topic | Study design |
|---|---|---|---|---|---|---|---|---|
| Sullivan J | 2021 | The Impact of Simulation-Based Education on Nurses' Perceived Predeployment Anxiety During the COVID-19 Pandemic Within the Cultural Context of a Middle Eastern Country. | Qatar | COVID-19 | Health workforce | Simulation | Impact | Before-after |
| Tegegne MD | 2022 | Use of social media for COVID-19-related information and associated factors among health professionals in Northwest Ethiopia: A cross-sectional study. | Ethiopia | COVID-19 | Health workforce | Virtual training | Knowledge evaluation | Cross-sectional |
| Jafree | 2022 | WhatsApp-Delivered Intervention for Continued Learning for Nurses in Pakistan During the COVID-19 Pandemic: Results of a Randomized-Controlled Trial. | Pakistan | COVID-19 | Health workforce | Virtual training | Knowledge evaluation | RCT |
| Leichner A; | 2021 | Mental health integration in primary health services after the earthquake in Nepal: a mixed-methods program evaluation. | Nepal | Earthquake | Health workforce and community | In-person training | Knowledge evaluation | Cross-sectional |
| Ng | 2009 | China-Australia training on psychosocial crisis intervention: response to the earthquake disaster in Sichuan. | China | Earthquake | Health workforce | In-person training | Impact | Before-after |
| AlAssaf | 2022 | Challenges in Pandemic Disaster Preparedness: Experience of a Saudi Academic Medical Center. | Saudi Arabia | COVID-19 | Health workforce | Blended | Emergency plans | Report |
| Button GJ | 2022 | Utilizing a "Crawl, Walk, Run" Training Model to Enhance Field Sanitation Capabilities for Peacekeeping Forces: A Recommendation for the Department of Defense Global Health Engagement Enterprise. | Senegal | COVID-19 | Military | Mixed | Learning technique | Report |
| Oliveira | 2020 | Personal Protective Equipment in the coronavirus pandemic: training with Rapid Cycle Deliberate Practice. | Brazil | COVID-19 | Health workforce | Simulation | Knowledge evaluation | Report |
| Morton Hamer MJ | 2019 | Enhancing Global Health Security: US Africa Command's Disaster Preparedness Program. | Africa | Disasters | Health workforce | In-person training | Emergency plans | Report |
| Khan JA | 2020 | Impact of multi-professional simulation-based training on perceptions of safety and preparedness among health workers caring for coronavirus disease 2019 patients in Pakistan. | Pakistan | COVID-19 | Health workforce | Simulation | Knowledge evaluation | Before-after |
| Jordans MJ | 2012 | Evaluation of a brief training on mental health and psychosocial support in emergencies: a pre- and post-assessment in Nepal. | Nepal | disasters | Health workforce | In-person training | Knowledge evaluation | Before-after |
| Lubogo M | 2015 | Ebola virus disease outbreak; the role of field epidemiology training programme in the fight against the epidemic, Liberia, 2014. | Liberia | Ebola | Health workforce | In-person training | Learning technique | Report |

**Table 1.** (Continued)

| Author | Year | Title | Country | Disaster/ Emergency | Population | Learning Method | Major topic | Study design |
|---|---|---|---|---|---|---|---|---|
| Lin L; | 2014 | The public health system response to the 2008 Sichuan province earthquake: a literature review and interviews. | China | Earthquake | National Institutions | N/A | Emergency plans | Report |
| Xia | 2020 | Evaluating the effectiveness of a disaster preparedness nursing education program in Chengdu, China. | China | Disasters | Health workforce | In-person training | Knowledge evaluation | RCT |
| Bajow N; | 2022 | Assessment of the effectiveness of a course in major chemical incidents for front line health care providers: a pilot study from Saudi Arabia. | Saudi Arabia | others | Health workforce | Simulation | Learning technique | Before-after |
| Saghafinia M | 2009 | Effect of the rural rescue system on reducing the mortality rate of landmine victims: a prospective study in Ilam Province, Iran. | Iran | others | Health workforce | In-person training | Knowledge evaluation | Observational |
| He | 2021 | Practice in Information Technology Support for Fangcang Shelter Hospital during COVID-19 Epidemic in Wuhan, China. | China | COVID-19 | Health workforce | Virtual training | Impact | Report |
| Kenar | 2006 | Medical preparedness against chemical and biological incidents for the NATO Summit in Istanbul and lessons learned. | Turkey | Disasters | Health workforce | In-person training | Emergency plans | Report |
| Salita C | 2019 | Development, implementation, and evaluation of a lay responder disaster training package among school teachers in Angeles City, Philippines: using Witte's behavioral model. | Philippines | Disasters | Experts and volunteers | In-person training | Knowledge evaluation | Before-after |
| Tauxe RV; | 2011 | Rapid development and use of a nationwide training program for cholera management, Haiti, 2010. | Haiti | Cholera | Health workforce | In-person training | Learning technique | Before-after |
| Werdhani RA | 2022 | A COVID-19 self-isolation monitoring module for undergraduate medical students: Linking learning and service needs during the pandemic surge in Indonesia. | Indonesia | COVID-19 | Health workforce | Virtual training | Impact | Report |
| Alshiekhly U | 2015 | Facebook as a learning environment for teaching medical emergencies in dental practice. | Syria | others | Health workforce | Virtual training | Impact | Cross-sectional |
| Cai W; | 2022 | Doctor of Public Health-Crisis Management and COVID-19 Prevention and Control: A Case Study in China. | China | COVID-19 | Health workforce | In-person training | Emergency plans | Report |
| Hageman | 2016 | Infection Prevention and Control for Ebola in Health Care Settings—West Africa and United States. | Africa | Ebola | Health workforce | In-person training | Emergency plans | Report |
| Pang | 2009 | Pilot training program for developing disaster nursing competencies among undergraduate students in China. | China | Disasters | Health workforce | In-person training | Knowledge evaluation | Before-after |
| Brisebois | 2011 | The Role 3 Multinational Medical Unit at Kandahar Airfield 2005–2010. | Afghanistan | War | Health workforce | Simulation | Emergency plans | Report |

(*Continued*)

**Table 1.** (Continued)

| Author | Year | Title | Country | Disaster/ Emergency | Population | Learning Method | Major topic | Study design |
|--------|------|-------|---------|---------------------|------------|-----------------|-------------|--------------|
| Gertler M | 2018 | West Africa Ebola outbreak— immediate and hands-on formation: the pre-deployment training program for frontline aid workers of the German Red Cross, other aid organizations, and the German Armed Forces, Wuerzburg, Germany 2014/15 | Africa | Ebola | Health workforce | In-person training | Impact | Report |
| Alim S | 2015 | Evaluation of disaster preparedness training and disaster drill for nursing students. | Indonesia | Disasters | Health workforce | In-person training | Knowledge evaluation | Before-after |
| Boon | 2009 | The impact of a community-based pilot health education intervention for older people as caregivers of orphaned and sick children as a result of HIV and AIDS in South Africa. | South Africa | others | Citizens and affected population | In-person training | Impact | Report |
| Subedi | 2018 | The Health Sector Response to the 2015 Earthquake in Nepal. | Nepal | Earthquake | Health workforce | In-person training | Emergency plans | Report |
| Cooper | 2012 | Evaluating the efficacy of the AAP "pediatrics in disaster" course: the Chinese experience. | International | Disasters | Health workforce | N/A | Impact | Before-after |
| Evans | 2016 | Innovation in Graduate Education for Health Professionals in Humanitarian Emergencies. | International | Disasters | Academia | In-person training | Learning technique | Report |
| Silva | 2021 | Implementation of COVID-19 telemonitoring: repercussions in Nursing academic training. | Brazil | COVID-19 | Health workforce | Virtual training | Knowledge evaluation | Report |
| Aghababaeian | 2013 | A comparative study of the effect of triage training by role-playing and educational video on the knowledge and performance of emergency medical service staffs in Iran. | Iran | Disasters | Health workforce | Blended | Knowledge evaluation | RCT |
| Chiu | 2021 | Facing the Coronavirus Pandemic: An Integrated Continuing Education Program in Taiwan. | China | COVID-19 | Health workforce | Virtual training | Knowledge evaluation | Before-after |
| Haar | 2020 | Strong families: a new family skills training program for challenged and humanitarian settings: a single-arm intervention tested in Afghanistan. | Afghanistan | Disasters | Citizens and affected population | In-person training | Impact | Before-after |
| Shi | 2020 | A simulation training course for family medicine residents in China managing COVID-19. | China | COVID-19 | Health workforce | Simulation | Knowledge evaluation | Before-after |
| Rouzier | 2013 | Cholera vaccination in urban Haiti. | Haiti | Cholera | Health workforce and community | In-person training | Emergency plans | Report |
| Abbas | 2018 | Peers versus professional training of basic life support in Syria: a randomized controlled trial. | Syria | Disasters | Health workforce | In-person training | Impact | RCT |
| Finnegan | 2015 | Preparing British Military nurses to deliver nursing care on deployment. An Afghanistan study. | Afghanistan | Disasters | Military | In-person training | Emergency plans | Report |
| Schreiber | 2004 | Hospital preparedness for possible nonconventional casualties: an Israeli experience. | Israel | others | Health workforce | In-person training | Emergency plans | Report |

(*Continued*)

**Table 1.** (Continued)

| Author | Year | Title | Country | Disaster/ Emergency | Population | Learning Method | Major topic | Study design |
|---|---|---|---|---|---|---|---|---|
| Phillips GA | 2014 | Capacity building for emergency care: Training the first emergency specialists in Myanmar. | Myanmar | Disasters | Health workforce | Academic training | Learning technique | Report |
| Sun L | 2021 | Intervention Effect of Time Management Training on Nurses' Mental Health during the COVID-19 Epidemic. | China | COVID-19 | Health workforce | In-person training | Impact | Before-after |
| Hung KKC | 2021 | Health Workforce Development in Health Emergency and Disaster Risk Management: The Need for Evidence-Based Recommendations. | International | Disasters | Health workforce | N/A | Emergency plans | Review |
| Mosquera A | 2015 | U.S. Public Health Service Response to the 2014–2015 Ebola Epidemic in West Africa: A Nursing Perspective. | Africa | Ebola | Health workforce | Simulation | Learning technique | Report |
| Bajow NA | 2019 | A Basic Course in Humanitarian Health Emergency and Relief: A Pilot Study from Saudi Arabia. | Saudi Arabia | Disasters | Health workforce | In-person training | Knowledge evaluation | Before-after |
| Chan SS | 2010 | Development and evaluation of an undergraduate training course for developing International Council of Nurses disaster nursing competencies in China. | China | Disasters | Health workforce | Academic training | Knowledge evaluation | Before-after |
| Ikeda S | 2022 | Introduction to the Project for Strengthening the ASEAN Regional Capacity on Disaster Health Management (ARCH Project). | Japan | Disasters | Health workforce | Academic training | Emergency plans | Report |
| Iskanderani AI | 2021 | Artificial Intelligence and Medical Internet of Things Framework for Diagnosis of Coronavirus Suspected Cases. | China | COVID-19 | Health workforce | Virtual training | Learning technique | Report |
| Rehman H | 2020 | Effectiveness of basic training session regarding the awareness of Ebola virus disease among nurses of public tertiary care hospitals of Lahore. | Pakistan | Ebola | Health workforce | In-person training | Knowledge evaluation | Before-after |
| Díaz-Guio DA | 2020 | Cognitive load and performance of health care professionals in donning and doffing PPE before and after a simulation-based educational intervention and its implications during the COVID-19 pandemic for biosafety. | Colombia | COVID-19 | Health workforce | Simulation | Knowledge evaluation | Before-after |
| Bai HX | 2020 | Artificial Intelligence Augmentation of Radiologist Performance in Distinguishing COVID-19 from Pneumonia of Other Origin at Chest CT. | China | COVID-19 | Health workforce | Virtual training | Knowledge evaluation | Observational |
| Tan W; | 2020 | Whole-Process Emergency Training of Personal Protective Equipment Helps Healthcare Workers Against COVID-19: Design and Effect. | China | COVID-19 | Health workforce | Simulation | Knowledge evaluation | Before-after |
| Kimani D | 2022 | Adopting World Health Organization Multimodal Infection Prevention and Control Strategies to Respond to COVID-19, Kenya. | Kenya | COVID-19 | Health workforce | In-person training | Learning technique | Report |

(*Continued*)

**Table 1.** (Continued)

| Author | Year | Title | Country | Disaster/ Emergency | Population | Learning Method | Major topic | Study design |
|---|---|---|---|---|---|---|---|---|
| Hu X; | 2022 | Creation and application of war trauma treatment simulation software for first aid on the battlefield based on undeformed high-resolution sectional anatomical image (Chinese Visible Human dataset). | China | Disasters | Health workforce | Virtual training | Impact | Report |
| Bajow N | 2015 | Proposal for a community-based disaster management curriculum for medical school undergraduates in Saudi Arabia | Saudi Arabia | Disasters | Health workforce | Academic training | Learning technique | Report |
| Kesavadev J; | 2021 | A new interventional home care model for COVID management: Virtual Covid IP. | India | COVID-19 | Health workforce | Virtual training | Learning technique | Observational |
| Sohn VY; | 2007 | From the combat medic to the forward surgical team: the Madigan model for improving trauma readiness of brigade combat teams fighting the Global War on Terror. | Iraq | War | Military | Simulation | Knowledge evaluation | Report |
| Cerqueira-Silva T | 2021 | Bridging Learning in Medicine and Citizenship During the COVID-19 Pandemic: A Telehealth-Based Case Study. | Brazil | COVID-19 | Health workforce | Virtual training | Impact | Report |
| Barbier O | 2018 | Has Current French Training for Military Orthopedic Surgeons Deployed in External Operations Been Appropriately Adapted? | Africa and Afghanistan | War | Military | Academic training | Learning technique | Observational |
| Leochico CFD; | 2021 | Role of Telerehabilitation in the Rehabilitation Medicine Training Program of a COVID-19 Referral Center in a Developing Country. | Philippines | COVID-19 | Health workforce | Virtual training | Learning technique | Report |
| Pereira BM | 2010 | Predeployment mass casualty and clinical trauma training for US Army forward surgical teams. | Iraq and Afghanistan | war | Military | In-person training | Impact | Report |
| Philip S | 2022 | A report on successful introduction of tele mental health training for primary care doctors during the COVID 19 pandemic. | India | COVID-19 | Health workforce | Virtual training | Knowledge evaluation | Report |
| Brearley MB | 2016 | Pre-deployment Heat Acclimatization Guidelines for Disaster Responders. | Philippines | Disasters | Military | Virtual training | Knowledge evaluation | Report |
| Das A; | 2022 | Implementation of infection prevention and control practices in an upcoming COVID-19 hospital in India: An opportunity not missed. | India | COVID-19 | Health workforce | Blended | Knowledge evaluation | Observational |
| Gareev I; | 2021 | The opportunities and challenges of telemedicine during COVID-19 pandemic. | International | COVID-19 | Health workforce | Virtual training | Learning technique | Report |
| Jensen | 2015 | Integration of Surgical Residency Training With US Military Humanitarian Missions. | South Asia | War | Military | In-person training | Knowledge evaluation | Report |
| Choufani | 2021 | Evaluation of a fellowship abroad as part of the initial training of the French military surgeon. | Africa | War | Military | In-person training | Impact | Report |
| Tashkandi | 2021 | Nursing strategic pillars to enhance nursing preparedness and response to COVID-19 pandemic at a tertiary care hospital in Saudi Arabia | Saudi Arabia | COVID-19 | Health workforce | In-person training | Knowledge evaluation | Report |

(*Continued*)

**Table 1.** (Continued)

| Author | Year | Title | Country | Disaster/Emergency | Population | Learning Method | Major topic | Study design |
|---|---|---|---|---|---|---|---|---|
| Chiu | 2021 | Developing and Implementing a Dedicated Prone Positioning Team for Mechanically Ventilated ARDS Patients During the COVID-19 Crisis. | China | COVID-19 | Health workforce | In-person training | Knowledge evaluation | Report |
| Operario | 2016 | Effect of a knowledge-based and skills-based programme for physicians on risk of sexually transmitted reinfections among high-risk patients in China: a cluster randomised trial. | China | others | Health workforce | In-person training | Knowledge evaluation | RCT |
| Wang | 2009 | Intervention to train physicians in rural China on HIV/STI knowledge and risk reduction counseling: preliminary findings. | China | others | Health workforce | In-person training | Knowledge evaluation | Before-after |
| El-Bahnasawy | 2015 | TRAINING PROGRAM FOR NURSING STAFF REGARDING VIRAL HEMORRHAGIC FEVERS IN A MILITARY HOSPITAL. | Egypt | others | Health workforce | In-person training | Impact | Before-after |
| Khari | 2022 | The Effect of E-Learning Program for COVID-19 Patient Care on the Knowledge of Nursing Students: A Quasi-Experimental Study. | Iran | COVID-19 | Health workforce | Virtual training | Knowledge evaluation | Before-after |
| Ripoll-Gallardo | 2020 | Residents working with Médecins Sans Frontières: training and pilot evaluation. | Africa | Disasters | Health workforce | Blended | Impact | Report |
| Otu | 2021 | Training health workers at scale in Nigeria to fight COVID-19 using the InStrat COVID-19 tutorial app: an e-health interventional study. | Nigeria | COVID-19 | Health workforce | Virtual training | Knowledge evaluation | Before-after |
| Lopes | 2020 | Adult learning and education as a tool to contain pandemics: The COVID-19 experience. | Africa | COVID-19 | Citizens and affected population | In-person training | Emergency plans | Opinion |
| Jackson | 2022 | Developing and Implementing Noninvasive Ventilator Training in Haiti during the COVID-19 Pandemic. | Haiti | COVID-19 | Health workforce | In-person training | Knowledge evaluation | Before-after |
| Sharma | 2021 | Effectiveness of Video-Based Online Training for Health Care Workers to Prevent COVID-19 Infection: An Experience at a Tertiary Care Level Institute, Uttarakhand, India. | India | COVID-19 | Health workforce | Virtual training | Knowledge evaluation | Before-after |
| Daniel | 2020 | Responding to Palliative Care Training Needs in the Coronavirus Disease 2019 Era: The Context and Process of Developing and Disseminating Training Resources and Guidance for Low- and Middle-Income Countries from Kerala, South India. | India | COVID-19 | Health workforce | Virtual training | Emergency plans | Report |
| Downie | 2022 | Remote Consulting in Primary Health Care in Low- and Middle-Income Countries: Feasibility Study of an Online Training Program to Support Care Delivery During the COVID-19 Pandemic. | Tanzania | COVID-19 | Health workforce | Virtual training | Knowledge evaluation | Report |

(*Continued*)

**Table 1.** (Continued)

| Author | Year | Title | Country | Disaster/ Emergency | Population | Learning Method | Major topic | Study design |
|---|---|---|---|---|---|---|---|---|
| Scott | 2020 | Training the Addiction Treatment Workforce in HIV Endemic Regions: An Overview of the South Africa HIV Addiction Technology Transfer Center Initiative. | South Africa | others | Experts and volunteers | In-person training | Knowledge evaluation | Before-after |
| Usami | 2018 | Addressing challenges in children's mental health in disaster-affected areas in Japan and the Philippines—highlights of the training program by the National Center for Global Health and Medicine. | Japan | Earthquake | Health workforce | In-person training | Impact | Report |
| Buyego | 2021 | Feasibility of Virtual Reality based Training for Optimising COVID-19 Case Handling in Uganda. | Uganda | COVID-19 | Health workforce | Virtual training | Impact | Report |
| Babu | 2021 | Simulated Patient Environment: A Training Tool for Healthcare Professionals in COVID-19 Era. | India | COVID-19 | Health workforce | Simulation | Knowledge evaluation | Before-after |
| Liu | 2022 | Development and Evaluation of Innovative and Practical Table-top Exercises Based on a Real Mass-Casualty Incident. | China | Disasters | Health workforce | Virtual training | Impact | Before-after |
| Fuenfer | 2009 | The U.S. military wartime pediatric trauma mission: how surgeons and pediatricians are adapting the system to address the need. | Afganistan and Iraq | War | Health workforce | Virtual training | Emergency plans | Report |
| Macht | 2022 | COVID-19: Development and implementation of a video-conference-based educational concept to improve the hygiene skills of health and nursing professionals in the Republic of Kosovo. | Kosovo | COVID-19 | Health workforce | Virtual training | Knowledge evaluation | Report |
| Jobson | 2019 | Targeted mentoring for human immunodeficiency virus programme support in South Africa | South Africa | others | Health workforce | In-person training | Impact | Report |
| Zelnick | 2018 | Training social workers to enhance patient-centered care for drug-resistant TB-HIV in South Africa. | South Africa | others | Health workforce | In-person training | Knowledge evaluation | Report |
| Cena-Navarro | 2022 | Biosafety Capacity Building During the COVID-19 Pandemic: Results, Insights, and Lessons Learned from an Online Approach in the Philippines. | Philippines | COVID-19 | Health workforce | Virtual training | Learning technique | Report |
| Richard | 2009 | Essential trauma management training: addressing service delivery needs in active conflict zones in eastern Myanmar. | Myanmar | War | Health workforce | In-person training | Knowledge evaluation | Report |
| Galagan | 2017 | Improving Tuberculosis (TB) and Human Immunodeficiency Virus (HIV) Treatment Monitoring in South Africa: Evaluation of an Advanced TB/HIV Course for Healthcare Workers. | South Africa | others | Health workforce | In-person training | Knowledge evaluation | Before-after |

(*Continued*)

**Table 1.** (Continued)

| Author | Year | Title | Country | Disaster/ Emergency | Population | Learning Method | Major topic | Study design |
|---|---|---|---|---|---|---|---|---|
| Irizarry | 2012 | Advanced Medical Technology Capacity Building and the Medical Mentoring Event: A Unique Application of SOF Counterinsurgency Medical Engagement Strategies. | Afghanistan | War | Health workforce and community | Simulation | Emergency plans | Report |
| Wanjiku | 2022 | Feasibility of project ECHO telementoring to build capacity among non-specialist emergency care providers. | Kenya | Disasters | Health workforce | Virtual training | Emergency plans | Report |
| Arnold | 2020 | Bridging the Gap Between Emergency Response and Health Systems Strengthening: The Role of Improvement Teams in Integrating Zika Counseling in Family Planning Services in Honduras. | Honduras | others | Health workforce | In-person training | Emergency plans | Before-after |
| Martinez | 2018 | Tourniquet Training Program Assessed by a New Performance Score. | Africa | War | Health workforce | In-person training | Knowledge evaluation | RCT |
| Kulshreshtha P | 2022 | Preparedness of Undergraduate Medical Students to Combat COVID-19: A Tertiary Care Experience on the Effectiveness and Efficiency of a Training Program and Future Prospects. | India | COVID-19 | Health workforce | Simulation | Knowledge evaluation | Before-after |
| Ahluwalia | 2021 | Effectiveness of remote practical boards and telesimulation for the evaluation of emergency medicine trainees in India. | India | Disasters | Health workforce | Virtual training | Impact | Report |
| Al-Hadidi | 2021 | Homemade cardiac and vein cannulation ultrasound phantoms for trauma management training in resource-limited settings. | Syria | Disasters | Health workforce | Simulation | Impact | Report |
| Patel | 2020 | "Emerging Technologies and Medical Countermeasures to Chemical, Biological, Radiological, and Nuclear (CBRN) Agents in East Ukraine" | Ukraine | War | Health workforce | In-person training | Emergency plans | Report |
| Wang | 2022 | Practical COVID-19 Prevention Training for Obstetrics and Gynecology Residents Based on the Conceive-Design-Implement-Operate Framework. | China | COVID-19 | Health workforce | Academic training | Impact | RCT |
| de Lesquen | 2020 | Adding the Capacity for an Intensive Care Unit Dedicated to COVID 19, Preserving the Operational Capability of a French Golden Hour Offset Surgical Team in Sahel. | Niger and Mali | COVID-19 | Military | Academic training | Emergency plans | Report |
| Ye | 2021 | Point-of-care training program on COVID-19 infection prevention and control for pediatric healthcare workers: a multicenter, cross-sectional questionnaire survey in Shanghai, China. | China | COVID-19 | Health workforce | Virtual training | Impact | Before-after |

(*Continued*)

**Table 1.** (Continued)

| Author | Year | Title | Country | Disaster/ Emergency | Population | Learning Method | Major topic | Study design |
|---|---|---|---|---|---|---|---|---|
| Bhattacharya | 2020 | Impact of a training program on disaster preparedness among paramedic students of a tertiary care hospital of North India: A single-group, before-after intervention study. | India | Disasters | Health workforce | Blended | Impact | Before-after |
| Wong | 2022 | ECMO simulation training during a worldwide pandemic: The role of ECMO telesimulation. | China | COVID-19 | Health workforce | Virtual simulation | Impact | Before-after |
| Fernández-Miranda | 2021 | Developing a Training Web Application for Improving the COVID-19 Diagnostic Accuracy on Chest X-ray. | Chile | COVID-19 | Health workforce | Virtual training | Knowledge evaluation | Report |
| Khoshnudi | 2022 | Comparison of the effect of bioterrorism education through two methods of lecture and booklet on the knowledge and attitude of nurses of Shams Al-Shomus Nezaja Hospital. | Iran | Disasters | Health workforce | In-person training | Knowledge evaluation | Before-after |
| Liesveld | 2022 | Teaching disaster preparedness to pre-licensure students: A collaborative project during the pandemic. | International | COVID-19 | Health workforce | Virtual training | Impact | Report |
| Shilkofski | 2017 | Pediatric Emergency Care in Disaster-Affected Areas: A Firsthand Perspective after Typhoons Bopha and Haiyan in the Philippines. | Philippines | Disasters | Health workforce | N/A | Emergency plans | Review |
| Bankole | 2021 | KNOWLEDGE OF HEALTH WORKERS ON CHOLERA MANAGEMENT IN OYO STATE: RESULTS OF A TRAINING INTERVENTION. | Nigeria | Cholera | Health workforce | In-person training | Knowledge evaluation | Before-after |
| | 2023 | The Use of Open-Source Online Course Content for Training in Public Health Emergencies: Mixed Methods Case Study of a COVID-19 Course Series for Health Professionals | International | COVID-19 | Health workforce | Virtual training | Learning technique | Qualitative |
| Klomp | 2020 | CDC's Multiple Approaches to Safeguard the Health, Safety, and Resilience of Ebola Responders | Africa | Ebola | Health workforce | In-person training | Emergency plans | Report |
| Van Hulle | 2020 | Tailoring malaria routine activities within the covid-19 pandemic: A risk and mitigation assessment of eight countries in West Africa | West Africa | others | Health workforce and community | Virtual training | Emergency plans | Report |
| Guerrero-Torres | 2020 | Impact of Training Residents to Improve HIV Screening in a Teaching Hospital in Mexico City | Mexico | others | Health workforce | In-person training | Knowledge evaluation | Before-after |
| Sena | 2020 | Disaster preparedness training in emergency medicine residents using a tabletop exercise | International | Disasters | Health workforce | In-person training | Learning technique | Before-after |
| Mitchell | 2020 | A partnership to develop disaster simulations for nursing students in response to climate change: description of a programme in Bluefields, Nicaragua, and Virginia, USA | Nicaragua y USA | Disasters | Health workforce | In-person training | Learning technique | Report |
| Amini | 2019 | Epidemiological profile of crimean congo hemorrhagic fever in afghanistan: A teaching-case study | Africa | others | Health workforce | In-person training | Learning technique | Report |

*(Continued)*

**Table 1.** (Continued)

| Author | Year | Title | Country | Disaster/ Emergency | Population | Learning Method | Major topic | Study design |
|--------|------|-------|---------|---------------------|------------|-----------------|-------------|--------------|
| Al-Mayahi | 2019 | Surveillance gaps analysis and impact of the late detection of the first middle east respiratory syndrome case in south batinah, oman: A teaching case-study | Africa | others | Health workforce | In-person training | Learning technique | Report |
| Pelican | 2019 | Building an Ebola-Ready workforce: Lessons learned on strengthening the global workforce through university networks | International | Ebola | Health workforce | In-person training | Emergency plans | Report |
| Peters | 2019 | Development and pilot testing of an infection prevention and control (IPC) tool for humanitarian response to outbreaks and natural disasters | Zambia | others | Health workforce | In-person training | Emergency plans | Before-after |
| Keita | 2018 | Impact of infection prevention and control training on health facilities during the Ebola virus disease outbreak in Guinea | West Africa | Ebola | Health workforce | In-person training | Knowledge evaluation | Observational |
| Edem-Hotah | 2018 | Utilizing Nurses to Staff an Ebola Vaccine Clinical Trial in Sierra Leone during the Ebola Outbreak | Sierra Leone | Ebola | Health workforce | In-person training | Emergency plans | Report |
| Mbanjumucyo | 2018 | Major incident simulation in Rwanda: A report of two exercises | Rwanda | Disasters | Health workforce | Simulation | Emergency plans | Report |
| Bruning M.D | 2018 | The kid next door-raising awareness of Civilian health care providers to the needs of military children-a tactical approach to education | Iraq and Afganistan | War | Health workforce | In-person training | Knowledge evaluation | Before-after |
| Andrews R | 2018 | Computer-assisted disaster response: Benefits for global healthcare | Africa | Disasters | Health workforce | In-person training | Emergency plans | Report |
| Githuku | 2017 | Cholera outbreak in homa bay county, kenya, 2015 | Africa | Cholera | Health workforce | In-person training | Learning technique | Report |
| Jones-Konneh | 2017 | Intensive education of health care workers improves the outcome of ebola virus disease: Lessons learned from the 2014 outbreak in Sierra Leone | Sierra Leone | Ebola | Health workforce | In-person training- simulation | Impact | Report |
| Phaup | 2017 | Increasing access to HIV treatment and care services for key populations in zambia: A partnership approach to strengthening local capacity to provide sensitivity training to health workers | Zambia | others | Health workforce | In-person training | Emergency plans | Report |
| Nwandu | 2017 | Sustainable pepfar funded in service HIV training delivery models: A training impact evaluation from nigeria | Nigeria | others | Health workforce | In-person training | Knowledge evaluation | Before-after |
| Laudisoit | 2017 | A One Health team to improve Monkeypox virus outbreak response: An example from the Democratic Republic of the Congo | Democratic Republic of Congo | others | Health workforce | In-person training | Emergency plans | Report |
| Umar | 2017 | Learningthroughservice: 'shifa homes'a project of Shifa College of Medicine for rehabilitation of flood victims | Pakistan | Disasters | Citizens and affected population | In-person training | Emergency plans | Report |
| Vaz | 2016 | The role of the polio program infrastructure in response to Ebola virus disease outbreak in Nigeria 2014 | Nigeria | Ebola | National Institutions | In-person training | Emergency plans | Report |

(*Continued*)

**Table 1.** (Continued)

| Author | Year | Title | Country | Disaster/ Emergency | Population | Learning Method | Major topic | Study design |
|---|---|---|---|---|---|---|---|---|
| Houben | 2016 | TIME Impact—a new user-friendly tuberculosis (TB) model to inform TB policy decisions | International | others | National Institutions | Virtual training | Emergency plans | Report |
| Umoren | 2016 | From global to local: Virtual environments for global-public health education | International | Disasters | Citizens and affected population | Virtual training | Emergency plans | Report |
| Garde | 2016 | Implementation of the first dedicated Ebola screening and isolation for maternity patients in Sierra Leone | Sierra Leone | Ebola | Health workforce | In-person training | Impact | Report |
| Toda | 2016 | The impact of a SMS-based disease outbreak alert system (mSOS) in Kenya | Kenya | others | Health workforce | In-person training | Impact | RCT |
| Shao X | 2015 | Evaluation of anti-Ebola training system in the Medical Team to Liberia and some suggestion | China | Ebola | Military | Blended | Emergency plans | Report |
| Ali | 2015 | Applicability of the advanced disaster medical response (ADMR) course, Trinidad and Tobago | Trinidad y Tobago | Disasters | Health workforce | In-person training | Learning technique | Before-after |
| Livingston | 2015 | Healthcare capacity building in Haiti: Training healthcare and non-healthcare providers in basic cardiopulmonary resuscitation | Haiti | Disasters | Health workforce and community | Simulation | Learning technique | Report |
| Berry | 2015 | How to set up an Ebola isolation unit: Lessons learned from Rokupa | Sierra Leone | Ebola | Health workforce | In-person training | Impact | Report |
| El-Bahnasawy | 2015 | Mosquito borne West Nile virus infection as a major threat | Egypt | others | Health workforce | In-person training | Impact | Report |
| Ariel | 2014 | The birth of family therapists: The kosova systemic family therapy training program | Kosovo | War | Health workforce | In-person training | Learning technique | Report |
| Reynolds | 2014 | Training health workers for enhanced monkeypox surveillance, Democratic Republic of the Congo | Congo | others | Health workforce | In-person training | Impact | Before-after |
| Acevedo | 2013 | Organization of the health system response to the 2009 H1N1 influenza pandemic in a hospital in Lima, Peru | Peru | others | Health workforce | In-person training | Emergency plans | Report |
| Hasanovic | 2013 | EMDR training for bosnia-herzegovina mental health workers in sarajevo as continuity of the building of psychotherapy capacity aftermath the 1992–1995 war | Bosnia-herzegovina | War | Health workforce | virtual training | Impact | Report |
| Hasanovic | 2013 | Training of bosnia-herzegovina mental health professionals in group analysis as the factor of development of culture of dialogue in the aftermath of the 1992–1995 war | Bosnia-herzegovina | War | Health workforce | In-person training | Impact | Report |
| Diaz | 2013 | Development of a severe influenza critical care curriculum and training materials for resource-limited settings | International | others | Health workforce | In-person training | Learning technique | Report |
| Asgary | 2013 | Comprehensive on-site medical and public health training for local medical practitioners in a refugee setting | Africa | Disasters | Health workforce | In-person training | Knowledge evaluation | Before-after |
| Plani | 2012 | Development of a hospital disaster plan and training exercises for chris hani baragwanath academic hospital and resource-limited countries | South Africa | Disasters | Health workforce | In-person training | Learning technique | Report |

(*Continued*)

**Table 1.** (Continued)

| Author | Year | Title | Country | Disaster/ Emergency | Population | Learning Method | Major topic | Study design |
|---|---|---|---|---|---|---|---|---|
| Wurapa | 2012 | Establishing a tropical medicine training program for the us department of defense (DOD) in kintampo, ghana: Overcoming challenges | Ghana | Disasters | Health workforce | In-person training | Emergency plans | Report |
| Chihanga | 2012 | Toward malaria elimination in Botswana: A pilot study to improve malaria diagnosis and surveillance using mobile technology | Botswana | others | Health workforce | virtual training | Learning technique | Report |
| Garnett | 2012 | Using south-south collaboration to strengthen midwifery skills and competencies in South Sudan | South Sudan | War | Health workforce | In-person training | Emergency plans | Report |
| Grosso | 2012 | The role of the international anaesthetist in the professional training and management of the anaesthesia nurses. 12 years of experience of emergency Italian NGO in Afghanistan | Afghanistan | War | Health workforce | In-person training | Emergency plans | Report |
| Norton | 2012 | The power of immersion; Training health personnel for disaster humanitarian responses | International | Disasters | Health workforce | In-person training | Emergency plans | Report |
| He | 2011 | The urgent rehabilitation technique education program for Wenchuan earthquake victims | China | Earthquake | Health workforce | In-person training | Emergency plans | Report |
| Sadiwa | 2011 | Addressing developmental delays among African children in post-conflict areas: An E-health approach | sierra Leone | War | Experts and volunteers | virtual training | Learning technique | Report |
| Oliveira | 2011 | Esperience in confronting the H1N1 epidemy | Brasil | others | Health workforce | In-person training | Impact | Report |
| Mbabazi | 2011 | Phase 1 implementation of male circumcision as a comprehensive package of HIV prevention in Rwanda | Rwanda | others | Health workforce | In-person training | Emergency plans | Report |
| Darby | 2011 | Multi-modal training for adult ICU nurses caring for paediatric patients in a war zone | Afghanistan | War | Health workforce | In-person training | Knowledge evaluation | Report |
| Way | 2011 | A modality of disaster response: Cyclone nargis and psychological first aid | Burma | Disasters | Health workforce and community | In-person training | Emergency plans | Report |
| Hasanovic | 2011 | Emdr training for mental health therapists in postwar bosniaherzegovina who work with psycho-traumatized population for increasing their psychotherapy capacities | Bosnia-herzegovina | War | Health workforce | In-person training | Emergency plans | Report |
| Oleribe | 2010 | From strategy to action: The vital roles of trained field epidemiologists and laboratory management professionals in epidemic control and prevention in Tanzania | Tanzania | others | Health workforce | In-person training | Emergency plans | Report |
| Kuhls | 2009 | International disaster training: Advanced disaster life support (ADLS) improves Thai physician and nurse confidence to respond to mass casualty disasters | Thailand | Disasters | Health workforce | In-person training | Knowledge evaluation | Before-after |

(*Continued*)

**Table 1.** (Continued)

| Author | Year | Title | Country | Disaster/ Emergency | Population | Learning Method | Major topic | Study design |
|---|---|---|---|---|---|---|---|---|
| van der Walt | 2006 | The effect of a CPD training (educational) intervention on the level of HIV knowledge of pharmacists | South Africa | others | Health workforce | virtual training | Knowledge evaluation | RCT |
| Peltzer | 2006 | A controlled study of an HIV/AIDS/ STI/TB intervention with traditional healers in KwaZulu-Natal, South Africa | South Africa | others | Health workforce | In-person training | Knowledge evaluation | RCT |
| Wondmikun | 2005 | Successful coupling of community attachment of health science students with relief work for drought victims | Ethiopia | Disasters | Health workforce | In-person training | Impact | Report |
| Kabir | 2021 | Association between preference and e-learning readiness among the Bangladeshi female nursing students in the COVID-19 pandemic: a cross-sectional study | Bangladesh | COVID-19 | Health workforce | Virtual training | Learning technique | Cross-sectional |
| AlOsta | 2023 | Jordanian nursing students' engagement and satisfaction with e-learning during COVID-19 pandemic | Jordania | COVID-19 | Health workforce | Virtual training | Learning technique | Report |
| Severini | 2023 | How to incorporate telemedicine in medical residency: A Brazilian experience in pediatric emergency | Brasil | COVID-19 | Health workforce | in-person training | Learning technique | Before-after |
| Conyers | 2023 | Where There's a War, There's a Way: A Brief Report on Tactical Combat Casualty Care Training in a Multinational Environment | International | War | Military | In-person training | Emergency plans | Report |
| Farhat | 2022 | The educational outcomes of an online pilot workshop in emergencies | Middle-east | Disasters | Health workforce | Virtual training | Emergency plans | Before-after |
| Mitchell | 2023 | Multimodal learning for emergency department triage implementation: experiences from Papua New Guinea during the COVID-19 pandemic | Papua New Guinea | Disasters | Health workforce | Virtual training | Knowledge evaluation | Before-after |
| Wang | 2022 | Rapid virtual training and field deployment for COVID-19 surveillance officers: experiences from Ethiopia | Ethiopia | COVID-19 | Health workforce | Virtual training | Emergency plans | Report |
| Suresh | 2021 | Predeployment training of Army medics assigned to prehospital settings | International | War | Health workforce | N/A | Knowledge evaluation | Report |
| Popova | 2022 | EXPERIENCE IN ORGANIZING URGENT TRAINING FOR GI PROFESSIONALS ON EMERGENCY CARE FOR ABDOMINAL INJURIES DURING THE WAR IN UKRAINE | Ukraine | War | Health workforce | Virtual training | Emergency plans | Report |
| Beckmann | 2022 | Training of psychotherapists in post-conflict regions: A Community case study in the Kurdistan Region of Iraq | Iraq | War | Health workforce | In-person training | Emergency plans | Report |
| Susanti | 2022 | The Effect of Caring Training on the Implementation of Caring Behavior and Work Culture of Nurses in Providing Services to COVID-19 Patients in an Indonesia's National Referral Hospital | Indonesia | COVID-19 | Health workforce | In-person training | Impact | Before-after |

(*Continued*)

**Table 1.** (Continued)

| Author | Year | Title | Country | Disaster/ Emergency | Population | Learning Method | Major topic | Study design |
|--------|------|-------|---------|---------------------|------------|-----------------|-------------|--------------|
| Canavese | 2022 | Massive Open Online Courses as Strategies to Address Violence through the Training of Health and the Intersectoral Professionals in Brazil | Brasil | COVID-19 | Health workforce | Virtual training | Learning technique | Report |
| Han | 2022 | Effect Analysis of "Four-Step" Training and Assessment Tool in the Prevention and Control of COVID-19 | China | COVID-19 | Health workforce | Virtual training | Learning technique | Report |
| Wood | 2022 | Evaluation of virtual online delivery of United Nations Office on Drugs and Crime (UNODC) national training on novel psychoactive substances (NPS) to healthcare professionals in Mauritius and the Seychelles during the COVID-19 pandemic | Mauritius and the Seychelles | COVID-19 | Health workforce | Virtual training | Knowledge evaluation | Before-after |
| Kamal | 2022 | Virtual Training on IGRT: A Unique Initiative of a Private Cancer Center from a Developing Country with Regional Academic Collaboration during COVID-19 Pandemic | Bangladesh | COVID-19 | Health workforce | Virtual training | Knowledge evaluation | Report |
| Fernandes Canesin | 2022 | Use of an innovative humanized virtual digital interactive heart failure clinical cases training strategy for cardiologist in the covid 19 pandemic | Brazil- Portugal and US | COVID-19 | Health workforce | Virtual training | Learning technique | Observational |
| Toro | 2022 | A Simulated Hospital in a COVID-19 Pandemic Environment for Undergraduate Neurology Students | Colombia | COVID-19 | Health workforce | simulation | Knowledge evaluation | Observational |
| Wong | 2022 | Better Surgical Ward Round: Replicating Near-Peer Teaching (NPT) on a virtual international platform during the COVID-19 pandemic. | International | COVID-19 | Health workforce | Virtual training | Learning technique | Before-after |
| Ordonez Juarez | 2022 | Virtual Learning Environment for Surgery Residents in a Third Level Hospital at Mexico City, a Teaching Alternative | Mexico | COVID-19 | Health workforce | Virtual training | Knowledge evaluation | Report |
| Payne | 2022 | The Development and Evaluation of Online Home Palliative Training During COVID-19 Pandemic in South Africa | South Africa | COVID-19 | Health workforce | Virtual training | Impact | Report |
| Kumar | 2022 | Effectiveness of virtual versus inperson training of the FCCS course: A comparative study | Ghana, Nigeria, Augusta | COVID-19 | Health workforce | Virtual training | Knowledge evaluation | Observational |
| Kalayasiri | 2021 | Training of psychiatry and mental health in a low- and middle-income country: Experience from Thailand before and after COVID-19 outbreak | Thailand | COVID-19 | Health workforce | Virtual training | Emergency plans | Report |
| Tang | 2021 | Combat casualty care training of Chinese peacekeeping military doctors: An evaluation of effectiveness | China | War | Military | In-person training | Knowledge evaluation | Report |
| Siddiqui | 2021 | The impact of a "one day basic intensive care training program" on knowledge of non-intensivists during the COVID-19 pandemic | India | COVID-19 | Health workforce | In-person training | Knowledge evaluation | Before-after |

(*Continued*)

**Table 1.** (Continued)

| Author | Year | Title | Country | Disaster/ Emergency | Population | Learning Method | Major topic | Study design |
|---|---|---|---|---|---|---|---|---|
| Groninger | 2021 | Project ECHO palliative care: Impact of TELE-mentoring and teaching for healthcare providers working with rohingya refugees in Bangladesh | Bangladesh | War | Health workforce | Virtual training | Emergency plans | Report |
| Daniel | 2021 | Evaluation of the faculty experience in developing and delivering palliative care e-resource toolkit for COVID-19 for low and middle income countries (LMICS) | international | COVID-19 | Health workforce | Virtual training | Emergency plans | Before-after |
| Kharel | 2021 | Impact of a virtual COVID-19 trainer of trainers program implemented via an academic-humanitarian collaboration | international | COVID-19 | Health workforce | Virtual training | Learning technique | Before-after |
| Thakre | 2020 | Evaluation of effectiveness of Covid-19 training and assessment of anxiety among nurses of a tertiary health care center during the Corona Virus pandemic-an experimental study | India | COVID-19 | Health workforce | In-person training | Emergency plans | Report |
| Rishipathak | 2021 | Assessing the effectiveness of online teaching methodology among emergency medical professionals in Pune, India | India | COVID-19 | Health workforce | Virtual training | Knowledge evaluation | Report |
| Khoja | 2016 | Impact of simple conventional and Telehealth solutions on improving mental health in Afghanistan | Afghanistan | COVID-19 | Health workforce | Virtual training | Emergency plans | Report |
| Bernstein | 2022 | The Power of Connections: AAP COVID-19 ECHO Accelerates Responses During a Public Health Emergency | USA | COVID-19 | Health workforce | Virtual training | Learning technique | Report |
| Hunt | 2022 | Facilitating Real-Time, Multidirectional Learning for Clinicians in a Low-Evidence Pandemic Response | USA | COVID-19 | Health workforce | Virtual training | Impact | Report |
| Hunt | 2021 | Virtual Peer-to-Peer Learning to Enhance and Accelerate the Health System Response to COVID-19: The HHS ASPR Project ECHO COVID-19 Clinical Rounds Initiative | USA | COVID-19 | Health workforce | Virtual training | Impact | Report |
| Lingum | 2021 | Building Long-Term Care Staff Capacity During COVID-19 Through Just-in-Time Learning: Evaluation of a Modified ECHO Model | Canada | COVID-19 | Health workforce | Virtual training | Impact | Report |
| Stephens | 2022 | Adapting a Telehealth Network for Emergency COVID-19 Pandemic Response, 2020–2021 | International | COVID-19 | Health workforce | Virtual training | Impact | Report |
| Begay | 2021 | Strengthening Digital Health Technology Capacity in Navajo Communities to Help Counter the COVID-19 Pandemic | USA | COVID-19 | Health workforce | Virtual training | Emergency plans | Report |

264, 298, 308] suggested a professional approach (such as Masters degrees and postgraduate studies in universities) as a way of preparing healthcare workers for health emergencies.

Within the virtual modalities, studies describe the use of telemedicine [92, 93, 95, 97, 193, 284, 290], Massive Online Open Courses (MOOC) [29, 167, 209, 281, 297], social Media [16,

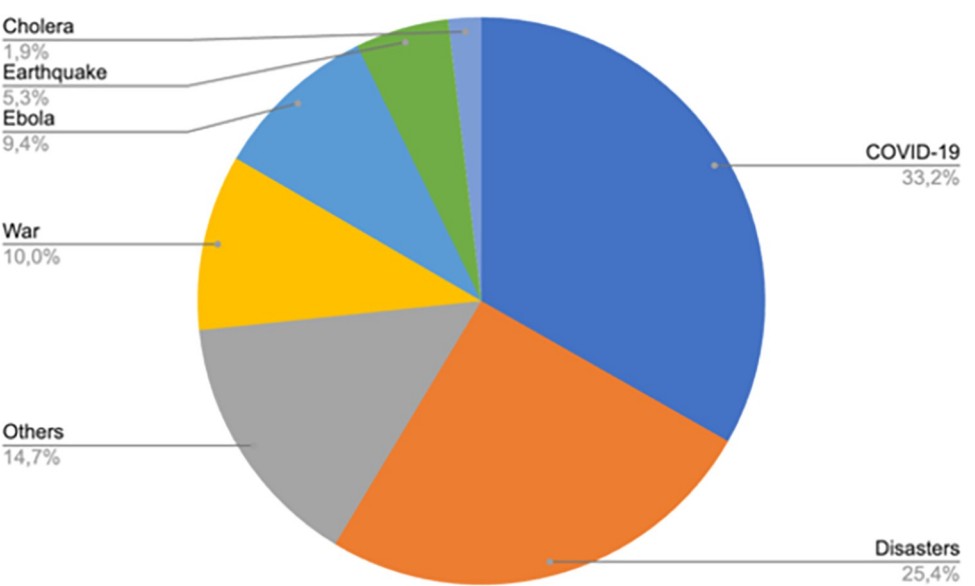

**Fig 2. Type of emergency distribution.**

266, 282, 285], gamification [184, 275], virtual simulation [89, 273], artificial intelligence [87, 301] and mobile devices [148, 203].

A few studies used blended format, usually workshops followed by virtual training (in-person and virtual) [36, 45, 55, 64, 65, 102, 127, 128, 153, 192, 242, 271, 302, 319–321] (Fig 3)

## Learners

Most of the learning interventions were directed towards the health workforce such as medical doctors, nurses, dentists, medical students, laboratory staff and paramedical students. Some

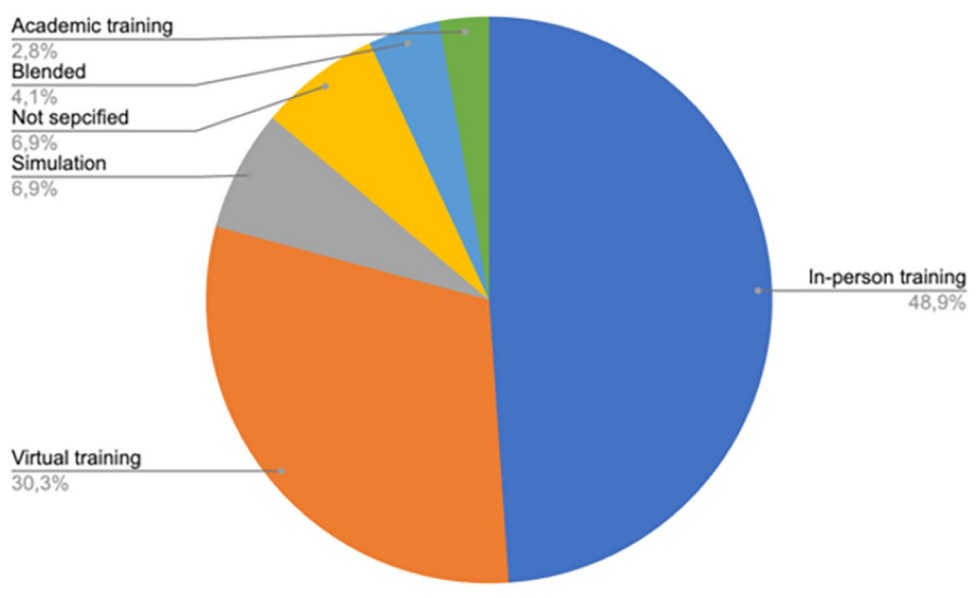

**Fig 3. Type of learning method distribution.**

learning interventions were also aimed at military personnel, citizens and affected population, volunteers, academia and national institutions. (S3 Table in S1 File)

## Learning content

Curricula content was mostly focused on disease and disaster management. Some content was about personal safety and security [49, 50, 55, 204–206, 208, 209], waste management [66, 212, 299], mental health [95, 105, 166, 172, 177, 213, 221, 235, 259, 277, 293], infection control and prevention [139, 168, 241, 264, 302], pest control [65, 66, 70, 212, 216, 299], triage [16, 62, 91, 94, 207, 271, 290] and stress management [31, 182, 270]. A smaller number of curricula included surgical training [98, 99, 165, 323], time management [223], stigma [269], language and local culture [20], humanitarian law and leadership [102], chemical incidents [214] and dealing with ethical dilemmas [72].

## Prominent topic areas

Four prominent topic areas were identified:

1. Knowledge acquisition: Articles that analyzed how much knowledge was acquired during learning interventions as a way to assess effectiveness of the intervention methods.

2. Emergency plans: Learning recommendations focusing on how governments and institutions should be prepared to face health emergencies.

3. Impact of the learning intervention: Articles that analyzed how learning and training during health emergencies had an impact on trainees and affected populations.

4. Training method: Articles that focus on different learning methods and tools that can be used for knowledge transfer during a health emergency.

The distribution of the prominent topic areas across the evidence is shown in Fig 4.

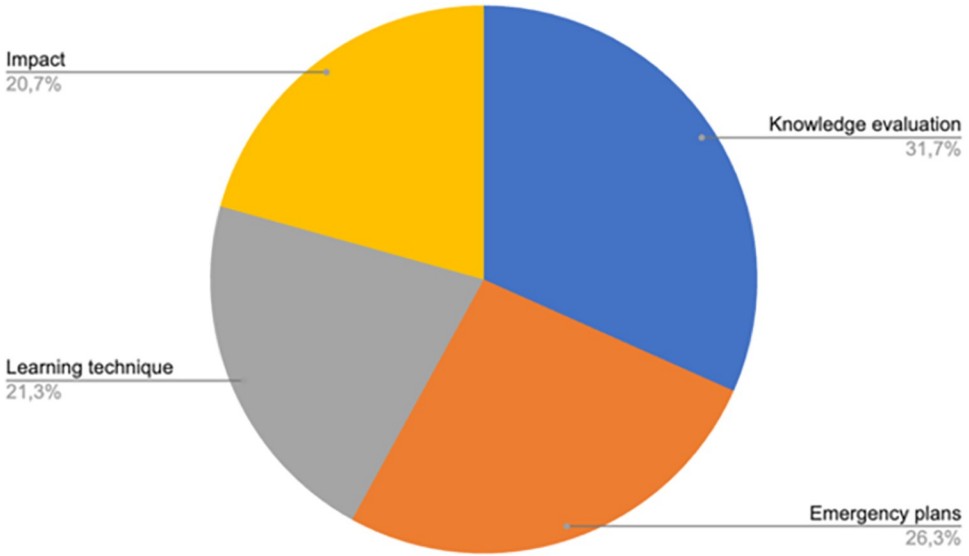

**Fig 4. Prominent topic areas distribution.**

The largest number of articles found (n = 101) correspond to "Knowledge Evaluation", followed by articles corresponding to "Emergency Plans" (n = 84) and "Training Method" (n = 68). Finally, "Impact" was analyzed in 66 studies.

## Knowledge acquisition

For knowledge evaluation, 101 studies [18, 19, 34, 42, 43, 49, 66, 80, 91, 95, 96, 98, 100, 101, 104, 108, 110, 112, 117, 156, 164, 169, 170, 173, 176, 178–181, 183, 186–188, 190, 195–199, 201, 203, 206–208, 212, 213, 215, 217, 218, 220, 222, 224–229, 231–236, 238, 240, 244–246, 249, 250, 253, 254, 257, 259, 260, 265–267, 269, 271, 273, 274, 279, 280, 284, 285, 287, 289–291, 293, 295, 299, 301–303, 305, 306, 316, 318] provided information about how a learning intervention during a health emergency can impact the knowledge of trainees. According to the results of these studies, learning interventions were helpful for improving knowledge, preparedness and confidence of trainees. Also, improved knowledge had a positive impact on diagnosis, and health outcomes such as rate of infected patients.

## Emergency plans

84 studies [26, 36, 38–40, 44, 50–52, 54, 55, 57, 60, 61, 63, 64, 67, 69, 71, 73–75, 78, 81–83, 86, 103, 107, 113, 114, 118, 119, 121, 122, 126–129, 131–135, 137, 142, 147, 149, 150, 152, 155, 157–159, 162, 163, 165, 166, 172, 174, 175, 177, 184, 191, 239, 248, 256, 263, 268, 275, 286, 288, 307–312, 317, 319, 320, 324–326] described the use of learning interventions as part of their emergency plans from countries and health institutions: 65 are reports, 6 are reviews and the rest of the evidence are randomized control trials, before and after studies, opinions, cross-sectional and qualitative studies. The evidence described how countries and hospitals should train healthcare workers, military, citizens and students in environmental emergencies (earthquake, tsunami, typhoons and disasters in general), biological emergencies (COVID-19, HIV, syphilis, Ebola, Cholera and Zika) and armed conflicts (war and chemical emergencies). The objective of this training was to prepare different populations for a future emergency and this explains why these interventions were not offered during an emergency.

As for the randomized controlled trials (RCT), Ma et al [275] compared the use of a gaming technique versus a disaster simulation and Lee et al [268] compared two video techniques: basic response to a fire versus generic volcanic emergency. In both studies, knowledge competence and response were statistically higher in the intervention groups (gaming technique and video response to fire emergencies).

## Training methods

On training methods, 68 studies [13–17, 20–25, 27–30, 32, 33, 37, 41, 45, 47, 48, 53, 58, 59, 62, 65, 68, 79, 84, 85, 87, 88, 90, 93, 97, 111, 123–125, 138, 141, 145, 146, 148, 153, 161, 167, 168, 170, 193, 204, 214, 216, 247, 251, 255, 261, 294, 297, 298, 300, 304, 313, 321, 322, 327] focused on describing different training methods that can be used during a health emergency. 15 studies described in-person training methods, and the rest of them used a virtual training method such as simulation, telemedicine, MOOC, videos or artificial intelligence.

MOOCs were used for cholera [29] and COVID-19 [167, 297, 315], and all of these studies showed that this method can be useful to disseminate trustworthy information in low- and middle-income countries and fill the gap of information during health emergencies.

Artificial intelligence [87] was used during COVID-19 to help radiologists to efficiently and timely diagnose suspected COVID-19 patients.

Videos [204] were implemented in Burkina Faso during a dengue epidemic. This study concluded that while videos are effective for knowledge transfer and training health professionals, the narrative genre of the videos can influence knowledge acquisition.

## Impact of learning interventions

Two studies measured impact as satisfaction in citizens and affected populations: One of them [262] trained parents of high-risk children about malaria prevention using an in-person technique. Participant satisfaction was assessed using a qualitative approach, however the main challenge reported was adherence to the course. The other study [185] reported on in-person training of Syrian refugee mothers of children with autism about overcoming trauma caused by war. Satisfaction was analyzed and improvements were recommended by the attendees, such as using online methods during the course.

In the rest of the evidence that reported satisfaction as an outcome, it was measured as self-perceived, as was mental health status of healthcare workers, well-being and preparedness.

27 studies analyzed how training during a health emergency impacted on the quality of life of healthcare workers, citizens and volunteers. The endpoints included: acceptability [296], anxiety [210], confidence [194], mental health [46, 200, 221, 223, 277, 314], preparedness [136, 140, 143, 144, 154, 160, 258, 270, 278, 328], satisfaction [31, 171, 199, 329–331], community cohesion [270] and social adaptation [292]. Most of the learning interventions were given after health emergencies occurred. This related to the fact that quality of life of the workers would be affected after the occurrence of these events. Two studies [194, 210], however, analyzed how quality of life could be improved by receiving prior training.

## Discussion

We found 319 studies that analyzed different learning interventions and training methods during health emergencies. Information about virtual training (including online platforms, artificial intelligence, MOOC, the use of mobile devices, social networks and telemedicine) arose mainly during the COVID-19 pandemic. Before the pandemic, most of the studies focused on in-person training. This change could be explained by technological advancements, the need for new information that had to be updated quickly and also because of social distancing recommendations. Of note, access to new technologies was scaled up after 2020 as the pandemic accelerated this process.

Many studies analyzed the impact of training on clinical outcomes such as patient survival [20, 216], rate of worker infection [34, 42, 88], number of hospitalizations [100, 300] and number of correct diagnoses [87, 301]. These findings will be important when developing guidance since these outcomes can be useful for decision making and selection of appropriate learning interventions and methods. Finally, the importance of coordinated work between different institutions [17, 26, 29, 39, 113, 239, 286], universities, governments and non-governmental organizations was noted in several studies. How best to foster coordinated work between institutions may be of great interest for future research.

We also found some potential evidence gaps that are worth investigating in future research studies. First, there is limited evidence on the content of learning interventions, especially regarding ethical dilemmas and how to solve them [72]. Also, we found a scarcity of evidence on end-of-life management during emergencies such as end-of-life care and managing deceased people. Subsequently, we identified other evidence gaps such as lack of robust evaluation of the effect of learning transfer, as most studies rely on self-reports (also referred to as post-training 'smile-sheets') to evaluate transfer of learning [332–334]. These studies provide limited actionable data to determine the effectiveness of training programs, as getting a

favorable reaction from learners does not guarantee that learning transfer has been achieved [335]. Further, too little research has examined the accountability of trainers for transfer in terms of using transfer-enhancing strategies, and how trainers are being evaluated [336]

In addition, most studies focused on information dissemination to build skills rather than employing evidence-informed training methods for performance improvement at the time of emergencies. Poor learning design and evaluation methods can result in learning being wasted [332]. Lastly, there was a lack of evidence on managing emergencies through teamwork, effective communication, and stress management. After all, factors such as stress, poor team dynamic due to hierarchical issues, and poor communication can lead to morbidity or mortality.

Overall, our scoping review on 'just in time' learning in crises situations dovetails with McGill and colleagues scoping review using the same methods to identify subsequent knowledge exchange processes in health crises [11].

## Limitations and strengths

This scoping review has potential limitations and strengths. One such limitation was the time constraint to deliver the scoping review quickly. Despite following standard scoping review processes, the search was not as robust as when performing a systematic review. However, once the *Learning in Health Emergencies* guidance is under way and new systematic reviews on the subject are commissioned, new evidence may arise to complement these findings. However, this scoping review fulfills a vital first step in establishing the amount and type of evidence from which subsequent systematic reviews were commissioned with subsequent critical and in-depth analysis. This work is the first to address this topic and will serve as a precursor for more comprehensive investigations. The scoping review followed an a priori protocol and was undertaken by a diverse team with varied expertise in the topic and scoping review methods.

## Conclusion

Research on learning and learning dissemination during health emergencies has revealed considerable advancements, particularly in virtual learning. Overall, it is evident that learning during health emergencies appears to improve knowledge, management, quality of life, satisfaction and clinical outcomes. All the information provided by this review can give decision makers tools to select different types of learning interventions and methods for healthcare workers, volunteers, military, civilians and governments. This scoping review may also be useful for future research to address the identified evidence gaps.

## Supporting information

**S1 File.**
(DOCX)

## Acknowledgments

The authors thank Selena Adams for reviewing this paper and the support of the Learning and Capacity Development Unit, the WHO Health Emergencies Programme and the World Health Organization.

## Author Contributions

**Conceptualization:** Heini Utunen, Elham Arabi, Anna Tokar.

**Data curation:** Giselle Balaciano.

**Formal analysis:** Giselle Balaciano, Jane Noyes.

**Funding acquisition:** Heini Utunen.

**Investigation:** Heini Utunen, Giselle Balaciano.

**Methodology:** Heini Utunen, Giselle Balaciano, Anna Tokar, Jane Noyes.

**Project administration:** Heini Utunen.

**Resources:** Heini Utunen.

**Supervision:** Heini Utunen, Anna Tokar.

**Validation:** Heini Utunen, Giselle Balaciano, Elham Arabi.

**Visualization:** Heini Utunen, Giselle Balaciano, Elham Arabi, Jane Noyes.

**Writing – original draft:** Heini Utunen, Giselle Balaciano, Elham Arabi, Anna Tokar, Jane Noyes.

**Writing – review & editing:** Heini Utunen, Giselle Balaciano, Elham Arabi, Anna Tokar, Aphaluck Bhatiasevi, Jane Noyes.

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
