## [Decision Letter · Decision Letter 0]

7 Sep 2023

PONE-D-23-24535Learning interventions and training methods in health emergencies: A scoping reviewPLOS ONE

Dear Dr. Balaciano,

Thank you for submitting your manuscript to PLOS ONE. After careful consideration, we feel that it has merit but does not fully meet PLOS ONE’s publication criteria as it currently stands. Therefore, we invite you to submit a revised version of the manuscript that addresses the points raised during the review process.

We look forward to receiving your revised manuscript.

Kind regards,

Nicholas Aderinto Oluwaseyi

Academic Editor

PLOS ONE

Journal Requirements:

Reviewers' comments:

Reviewer's Responses to Questions

**Comments to the Author**

1. Is the manuscript technically sound, and do the data support the conclusions?

Reviewer #1: Partly

Reviewer #2: Yes

2. Has the statistical analysis been performed appropriately and rigorously? 

Reviewer #1: N/A

Reviewer #2: N/A

3. Have the authors made all data underlying the findings in their manuscript fully available?

Reviewer #1: Yes

Reviewer #2: Yes

4. Is the manuscript presented in an intelligible fashion and written in standard English?

Reviewer #1: Yes

Reviewer #2: Yes

5. Review Comments to the Author

Reviewer #1: As per attached document.

The topic is very relevant, but the paper needs refinement and has to come across as more scientific in its approach, definitions and critical analysis. You have likely been limited by your databases, which is a flaw. Grey literature should have been used.

Reviewer #2: OVERVIEW

The paper was well written, and systematically designed, the authors sought to understand how learning interventions have evolved during pandemics with the aim of mapping available evidence and constructing recommendation and policy briefs to ensure a more appropriate and faster response during future emergencies.

FEW COMMENTS

1. The Discussion section is not satisfactory considering the large volume of available resources for discussion. I.e the article would benefit from a more robust discussion.

2. The authors didn't place much emphasis on techniques that failed or were not significantly successful so as to allow for either an improvement on such techniques or a total abandonment.

6. PLOS authors have the option to publish the peer review history of their article (what does this mean?). If published, this will include your full peer review and any attached files.

Reviewer #1: **Yes: **Kunal D Patel

Reviewer #2: **Yes: **Gbolahan Olatunji

---

## [Author Response · Author response to Decision Letter 0]

15 Nov 2023

Dear reviewer:

We appreciate your comments and very valuable inputs to our work. We hereby submit our responses below:

General Comments:

1) Do not discuss the big elephant in the room – lack of accreditation (i.e university level standard of training or royal college standard). WHO standard is only a guide and does not meet the QA of training that universities provide – surely, emergency training must be meet this standard? This can be addressed in the introduction and discussion. Would standards, e.g a term such as ‘accredited’ affected your search, and possibly made for a different, but more accurate paper? 

Answer: Our scoping review focuses mainly on just in time learning and professional accreditation is excluded. We further clarified inclusion and exclusion criteria in line 152. 

2) A weakness is that only 2 databases were used, for a scoping review more databases should have been utilized. Additionally, grey literature should have been considered, considering the WHO involvement. 

Answer: This scoping review included only peer reviewed literature. WHO has done a grey literature review and will be published in a separate article. We expanded the search to HINARI, Web of Science, WorldCat and CABI and updated the results accordingly.

3) In your methodology, you state you are ‘analyzing’ and ‘developing an understanding’, for this review to have merit, your analysis should be either narrative but ideally thematic in nature and this is not clear. Line 280 says ‘groupings’ which is not very scientific, your analysis structure needs to be clearer. 

Answer: To avoid confusion, we have removed mention of analyzing and grouping data and reverted to using Arskey and O’Malley’ exact language of collating/summarizing. We present the results as an evidence map. We then summarized the studies by major topic areas and briefly described key aspects of the studies under these headings. A recent similar scoping review using Arskey and O’Malley’s methodology publish by PLOS ONE can be found here: https://pubmed.ncbi.nlm.nih.gov/36827258/. Of note McGill’s scoping review included far fewer studies (86) compared with our review (319).

4) Major topic areas – you do not go into enough detail. You need to describe the papers well, with data, geography and numbers. It comes across as too broad and not specific in terms of analysis. 

Answer: This scoping review included a huge number of studies (319). A lot of detail at individual study level can be found in the included study table. The visual displays provide additional collated information. Within the limited scope of collating/summarizing of Arskey and O’Malley’s scoping review methodology, there is not a methodological requirement to describe the studies in detail and given the huge number of studies it would not have been feasible to do so. As stated in line 126 the objective of this scoping review is to map the evidence so that subsequently review questions can be identified and subsequent systematic reviews can be commissioned by WHO.

5) Table of papers should be included in the main body of the paper, not as a supplement. 

Answer: It has been moved to line 232. The table reporting 319 studies is however enormous. We welcome any feedback from the managing Editor as to whether this table is too big to include in the body of the main manuscript.

6) Change your referencing style if possible. 

Answer: Referencing style is Vancouver, suggested by the journal. If there is another referencing style we should use, please let us know and we will change it.

7) Your discussion, though touching on your findings, does not explore the literature in terms of the ‘major topic areas’ you found – what else is discussed in the literature around this? E.g MOOCs are considered poor educational tools, how does this compare with your findings? AI use, what is currently being pubished and found,, compared to your findings? This highlights a lack of critical analysis in the discussion and this needs to be addressed. 

Answer: As stated in line 126, the objective of this scoping review was to map the existing evidence on the topic. We decided to undertake a scoping review (and not a systematic review) in order to map 319 included studies. Scoping reviews are generally not designed to critically evaluate the content or results of included studies or situate what was found in the context of broader literature. 

Specific Comments:

1) Line 24: ‘the latest evolving knowledge’ – knowledge of what? Needs to be rewritten. 

Answer: We rewrote the paragraph 

2) Line 43: Abstract results should be more detailed in terms of types of papers used and in brackets original findings to what was analysed. 

Answer: The PLOS ONE abstract is limited to 300 words. According to the PLOS ONE author guidance, ‘The Abstract should be succinct; it must not exceed 300 words. Authors should mention the techniques used without going into methodological detail and should summarize the most important results without going into detail’. Our abstract is currently 325 words and we therefore need to remove words and further cut back on detail rather than add.

3) Line 60-72: More definition around what emergencies are and what is referred to as system strengthening – the reader should be able to understand why, though in general why certain systems need strengthening – whether this be due to lack of infrastructure or more importantly no UHC, with private healthcare hindering responses

Answer: We rewrote the introduction.

4) Line 89 – the ‘ongoing’ covid 19 pandemic, do not assume it is over or refer to it in the past tense Changed the verbal term to present

Answer: We rewrote the introduction.

5) Line 233: Location paragraph needs work: 3 countries only? Yet, your table has papers from Lebanon and other countries in sub Saharan Africa. This needs to be corrected and be more detailed. (Again due to the weakness of databases and search, most papers were from 3 geographies, China, India and South Africa. You mention this is due to covid yet, your search time frame was from 2003, this does not make sense and needs a re-evaluation in terms of papers and search.) 

Answer: We rewrote the paragraph in line 252 and added a figure

● Figure 3: percentages do not add up to 100%, instead 99.3 (strongly advised against the use of pie charts, bar charts are more appropriate). Plus legends of figures need to be more detailed

Answer: We corrected the figure, now it is figure 4.

---

## [Decision Letter · Decision Letter 1]

9 Feb 2024

PONE-D-23-24535R1Learning interventions and training methods in health emergencies: A scoping reviewPLOS ONE

Dear Dr. Balaciano,

Thank you for submitting your manuscript to PLOS ONE. After careful consideration, we feel that it has merit but does not fully meet PLOS ONE’s publication criteria as it currently stands. Therefore, we invite you to submit a revised version of the manuscript that addresses the points raised during the review process.

We look forward to receiving your revised manuscript.

Kind regards,

Nicholas Aderinto Oluwaseyi

Academic Editor

PLOS ONE

Reviewers' comments:

Reviewer's Responses to Questions

**Comments to the Author**

1. If the authors have adequately addressed your comments raised in a previous round of review and you feel that this manuscript is now acceptable for publication, you may indicate that here to bypass the “Comments to the Author” section, enter your conflict of interest statement in the “Confidential to Editor” section, and submit your "Accept" recommendation.

Reviewer #1: All comments have been addressed

Reviewer #2: All comments have been addressed

Reviewer #3: All comments have been addressed

2. Is the manuscript technically sound, and do the data support the conclusions?

Reviewer #1: Partly

Reviewer #2: Yes

Reviewer #3: Yes

3. Has the statistical analysis been performed appropriately and rigorously? 

Reviewer #1: N/A

Reviewer #2: N/A

Reviewer #3: Yes

4. Have the authors made all data underlying the findings in their manuscript fully available?

Reviewer #1: Yes

Reviewer #2: Yes

Reviewer #3: Yes

5. Is the manuscript presented in an intelligible fashion and written in standard English?

Reviewer #1: Yes

Reviewer #2: Yes

Reviewer #3: Yes

6. Review Comments to the Author

Reviewer #1: • The authors have demonstrated a good understanding of the reviewer comments and have responded. However, even now it is clear that this paper is lacking in critical analysis and discussion, making much less robust than it should be. For example, it is appreciated that over 300 papers were found. Considering the search terms, this is not a surprise. However, even at this large number an in depth analysis should have been done. Yes, Arksey and Malley, have provided a foundation but strong scoping reviews analyse the papers found and not simply describe them. It is very feasible to analyse these papers, which contradicts the response “huge number of studies it would not have been feasible to do so”.

• Figure 3 is unfortunately not of any use – pie charts in general are not ideal for this amount of data.

• Using a scoping review as 'limited in purpose' as a limitation, excuses the lack of any in depth analysis. All types of reviews have limitations, however some of the most successful and impactful scoping reviews have an in-depth analysis. This a real shame. The topic and paper overall is an interesting read, but, lack of proper analysis. This is reflected in the discussion.

Reviewer #2: The authors seem to have made the necessary changes. In my view, the article serves as a precursor to a more comprehensive project and should be regarded as such. It appears to be an extensive information gathering effort with moderate discussions on leveraging it for improved outcomes. Nevertheless, the article, as mentioned earlier, effectively paves the way for a more detailed project.

Reviewer #3: The Editorial Team, PLOS ONE

Thank you for inviting me to review this interesting article. The scoping review article focuses on comprehensively mapping learning interventions and training employed during health emergencies with the goal of influencing global policies and future research.

Because this submission has been previously reviewed, I also evaluated the adequacy of the authors’ response to previous review queries.

General Summary of Authors’ Response to Previous Review Queries

a. Accreditation and Standards: the authors clarified that the aim of the scoping review was to evaluate in time learning, excluding professional accreditation. This is in keeping with the study objectives

b. The authors also expanded the database search to provide a more robust coverage

c. The queries on the methodology and results have also been duly addressed.

Review of Current Manuscript

A. Title and Abstract

1. The title correctly captures the objectives of the study.

2. In their response to the reviewers comment 2, the authors stated they have expanded the search database. However, only two databases are mentioned in the methods section of the abstract (line 40-41). They authors should consider updating this.

B. Introduction:

The introduction effectively discusses the purpose of the study.

C. Methodology

1. The study employs a good methodology framework, including search strategies and data extraction.

2. The authors should consider including the rationale for excluding grey literature from their review in the body of the text

D. Results

The results section is well presented.

E. Discussion

The conclusion succinctly summarizes the findings of the study and provides a background for future research.

F. List of Abbreviations

The authors should consider including a comprehensive list of abbreviation

G. References

The references were according to the journal guideline

H. Table

The study includes a comprehensive table of findings.

7. PLOS authors have the option to publish the peer review history of their article (what does this mean?). If published, this will include your full peer review and any attached files.

Reviewer #1: **Yes: **Kunal D Patel

Reviewer #2: **Yes: **Gbolahan Olatunji

Reviewer #3: **Yes: **Ismaila Ajayi Yusuf

---

## [Author Response · Author response to Decision Letter 1]

8 Mar 2024

Dear reviewers:

We appreciate your comments and very valuable inputs to our work. We hereby submit our responses below. 

Major comment to the authors from the Reviewer 1:

Thank you to the three peer reviewers for reviewing the revised manuscript. We note that Reviewer 2 and Reviewer 3 are satisfied with the format, methods and content of the scoping review and the comments received have been actioned on from our side. These two reviewers conclude this scoping review reported in the manuscript follows a similar purpose and format to scoping reviews that we have previously published in this journal(1), and others have published in this journal using the same methodology(2). 

REVIEWER 1 COMMENT: 

Using a scoping review as 'limited in purpose' as a limitation, excuses the lack of any in depth analysis. All types of reviews have limitations, however some of the most successful and impactful scoping reviews have an in-depth analysis. This a real shame. The topic and paper overall is an interesting read, but, lack of proper analysis. This is reflected in the discussion.

RESPONSE: We acknowledge that reviewer 1 is requesting more clarifications to the manuscript including a critical analysis of the literature. We have addressed our concerns about critical analysis in the general comments and we have updated our discussion in line 460.

We have included three extracts here from recent seminal methodological works on the purpose of a scoping review as a sample no mention of a PRISMA for scoping reviews(3), beyond charting and describing, additional analyses are not required. Critical appraisal of the included sources has also been optional in other works(4, 5). We therefore suggest this manuscript to be processed as is, without the additional more-in-depth critical analysis of the literature and hope this is an agreeable way forward, in light of the referenced work and in line with the Reviewers 2 and 3 who are in favor of our manuscript. 

Detailed comments to the Authors 

COMMENT Figure 3 is unfortunately not of any use – pie charts in general are not ideal for this amount of data.

RESPONSE We have deleted figure number 3, all necessary information can be found in the summary of studies table.

COMMENT: Only two databases are mentioned in the methods section of the abstract (line 40-41). They authors should consider updating this.

RESPONSE This has been updated in line 42 by adding the full database search.

COMMENT The authors should consider including a comprehensive list of abbreviation

RESPONSE Abbreviations has been added following PLOS one guidelines.

COMMENT The authors should consider including the rationale for excluding grey literature from their review in the body of the text.

RESPONSE: This issue has been addressed in line 195 by adding the justification for the exclusion of grey literature from this paper.

References:

1. Silveira Bianchim M., Crane E., Jones A., Neukirchinger B., Roberts G., McLaughlin L., Noyes J. The implementation, use and impact of patient reported outcome measures in value-based healthcare programmes: A scoping review. PLoS One. 2023;18(12):e0290976.

2. McGill E., Halliday E., Egan M., Popay J. Knowledge exchange in crisis settings: A scoping review. PLoS One. 2023;18(2):e0282080.

3. Tricco A. C., Lillie E., Zarin W., O'Brien K. K., Colquhoun H., Levac D., Moher D., Peters M. D. J., Horsley T., Weeks L., Hempel S., Akl E. A., Chang C., McGowan J., Stewart L., Hartling L., Aldcroft A., Wilson M. G., Garritty C., Lewin S., et al. PRISMA Extension for Scoping Reviews (PRISMA-ScR): Checklist and Explanation. Ann Intern Med. 2018;169(7):467-73.

4. Peters M. D. J., Marnie C., Colquhoun H., Garritty C. M., Hempel S., Horsley T., Langlois E. V., Lillie E., O'Brien K. K., Tuncalp Ӧ, Wilson M. G., Zarin W., Tricco A. C. Scoping reviews: reinforcing and advancing the methodology and application. Syst Rev. 2021;10(1):263.

5. Munn Z., Peters M. D. J., Stern C., Tufanaru C., McArthur A., Aromataris E. Systematic review or scoping review? Guidance for authors when choosing between a systematic or scoping review approach. BMC Med Res Methodol. 2018;18(1):143.

---

## [Decision Letter · Decision Letter 2]

2 May 2024

Learning interventions and training methods in health emergencies: A scoping review

PONE-D-23-24535R2

Dear Dr. Balaciano,

We’re pleased to inform you that your manuscript has been judged scientifically suitable for publication and will be formally accepted for publication once it meets all outstanding technical requirements.

Kind regards,

Nicholas Aderinto Oluwaseyi

Academic Editor

PLOS ONE

Additional Editor Comments (optional):

Reviewers' comments:

Reviewer's Responses to Questions

**Comments to the Author**

1. If the authors have adequately addressed your comments raised in a previous round of review and you feel that this manuscript is now acceptable for publication, you may indicate that here to bypass the “Comments to the Author” section, enter your conflict of interest statement in the “Confidential to Editor” section, and submit your "Accept" recommendation.

Reviewer #1: All comments have been addressed

2. Is the manuscript technically sound, and do the data support the conclusions?

Reviewer #1: Yes

3. Has the statistical analysis been performed appropriately and rigorously? 

Reviewer #1: N/A

4. Have the authors made all data underlying the findings in their manuscript fully available?

Reviewer #1: Yes

5. Is the manuscript presented in an intelligible fashion and written in standard English?

Reviewer #1: Yes

6. Review Comments to the Author

Reviewer #1: Comments addressed. Please follow up with another paper as soon as possible, the world needs to see the analysis!

7. PLOS authors have the option to publish the peer review history of their article (what does this mean?). If published, this will include your full peer review and any attached files.

Reviewer #1: No

---

## [Editor Report · Acceptance letter]

14 Jun 2024

PONE-D-23-24535R2 

PLOS ONE

Dear Dr. Balaciano, 

I'm pleased to inform you that your manuscript has been deemed suitable for publication in PLOS ONE. Congratulations! Your manuscript is now being handed over to our production team.

Kind regards, 

on behalf of

Dr. Nicholas Aderinto Oluwaseyi 

Academic Editor

PLOS ONE